# Utility of polygenic scores across diverse diseases in a hospital cohort for predictive modeling

Ting-Hsuan Sun [1], Chia-Chun Wang[1], Ting-Yuan Liu [2], Shih-Chang Lo [1], Yi-Xuan Huang [1], Shang-Yu Chien[1], Yu-De Chu [1], Fuu-Jen Tsai [3,4,5,6,9] & Kai-Cheng Hsu [1,7,8,9]

Polygenic scores estimate genetic susceptibility to diseases. We systematically calculated polygenic scores across 457 phenotypes using genotyping array data from China Medical University Hospital. Logistic regression models assessed polygenic scores' ability to predict disease traits. The polygenic score model with the highest accuracy, based on maximal area under the receiver operating characteristic curve (AUC), is provided on the GeneAnaBase website of the hospital. Our findings indicate 49 phenotypes with AUC greater than 0.6, predominantly linked to endocrine and metabolic diseases. Notably, hyperplasia of the prostate exhibited the highest disease prediction ability ($P$ value $= 1.01 \times 10^{-19}$, AUC $= 0.874$), highlighting the potential of these polygenic scores in preventive medicine and diagnosis. This study offers a comprehensive evaluation of polygenic scores performance across diverse human traits, identifying promising applications for precision medicine and personalized healthcare, thereby inspiring further research and development in this field.

Significant progress in genetics has deepened our understanding of the genetic underpinnings of complex traits and diseases. A notable development is the creation and application of polygenic scores (PGSs), which predict an individual's risk for specific traits or diseases based on their genetic profile[1,2]. The theory of polygenic inheritance, which posits that traits or diseases result from the interaction of multiple genes, has been a topic of discussion for many years. However, it was the introduction of genome-wide association studies (GWASs) in the early 2000s that brought the concept of PGSs into broader use[3]. GWASs have empowered researchers to identify thousands of genetic variants linked to various traits and diseases by examining the entire genome of large populations.

The initial PGSs were computed using a straightforward approach known as the "burden test" or the single nucleotide polymorphism (SNP)-based method. This method involves tallying the total number of risk alleles (i.e., genetic variants associated with increased risk) for an individual across multiple loci to generate a composite score[4,5]. However, this approach does not consider the varying effect sizes of different genetic variants, resulting in scores with limited predictive accuracy. As a result, more sophisticated statistical methods, such as the "weighted method"[6] or linkage disequilibrium score regression[7], have been developed. These methods account for the effect sizes of different variants and the correlations between variants through the measurement of linkage disequilibrium. By incorporating additional

[1]Artificial Intelligence Center, China Medical University Hospital, Taichung 40447, Taiwan. [2]Million-person Precision Medicine Initiative, Department of Medical Research, China Medical University Hospital, Taichung 40447, Taiwan. [3]Department of Medical Research, China Medical University Hospital, Taichung 40447, Taiwan. [4]School of Chinese Medicine, China Medical University, Taichung 40402, Taiwan. [5]Division of Pediatric Genetics, Children's Hospital of China Medical University, Taichung 40447, Taiwan. [6]Department of Biotechnology and Bioinformatics, Asia University, Taichung 41354, Taiwan. [7]Department of Neurology, China Medical University Hospital, Taichung 40447, Taiwan. [8]Department of Medicine, China Medical University, Taichung 40402, Taiwan. [9]These authors contributed equally: Fuu-Jen Tsai, Kai-Cheng Hsu. e-mail: 000704@tool.caaumed.org.tw; kaichenghsu66@gmail.com

information, such as the effect sizes and frequencies of variants, these methods can produce accurate and robust PGSs[8,9].

PGSs have found extensive application in various traits, including height, body mass index[10], and intelligence[11], as well as diseases such as cardiovascular disease[12,13], cancer[14–16], and psychiatric disorders[17–19]. PGSs have facilitated investigations into the genetic foundations of complex traits and diseases, the identification of individuals at high risk for certain diseases and conditions[20], and the exploration of gene-environment interactions. The Polygenic Score Catalog (https://www.pgscatalog.org/)[21] was developed to streamline the distribution of PGSs. This catalog, adhering to standardized procedures for quality control, data curation, and metadata annotation, serves as a centralized resource enabling researchers and clinicians to access and utilize PGSs for various applications, including risk prediction, personalized medicine, and genetic research.

In this study, we procured SNP array data from a cohort of 276,712 individuals, whose data were stored in the electronic health record system of China Medical University Hospital (CMUH) in Taiwan. Utilizing PGS files from the PGS Catalog, which contain data on genetic variants and their corresponding weights, we calculated PGSs for 457 disease traits. We evaluated their predictive performance using logistic regression models. The results were subsequently stored in the CMUH GeneAnaBase, a platform that enables population health research and facilitates investigations into the genetic basis of diseases, novel genetic

associations, and heritability. This study offers valuable insights into the genetic similarities of diseases in Taiwan and contributes to our understanding of disease genetics in the context of population health.

## Results

### Distribution of performance metrics and ancestry cohorts

A comprehensive analysis was conducted on a total of 13,097 performance records available in the PGS Catalog, examining various conditions, including consideration of covariates and different ancestry cohorts (Fig. 1B). Among these records, three primary performance measurements were extracted, constituting 27.27% (3572 records) of the total. The most frequently utilized measurement was the AUC, comprising 2194 records, followed by the odds ratio, with 1513 records, and the hazard ratio, with 419 records. The remaining 73.73% (9657 records) utilized alternative calculation methods such as $R^2$, Nagelkerke's $R^2$, the z-test, and Youden's index (Fig. 1A).

Regarding the sample cohort used to develop PGS, a total of 2153 records were available. Since 60.94% (1312 records) of the data lacked case/control values, the number of individuals was used for statistical analysis. From the data distribution (Fig. 1D), it was observed that 50% of PGS were developed using samples of less than 23,072 individuals. Following an initial screening step, 507 PGS were retained, and the cumulative distribution plot (Fig. 1E) illustrates that 50% of PGS used sample sizes larger than 269,704. This highlights a trend in our process

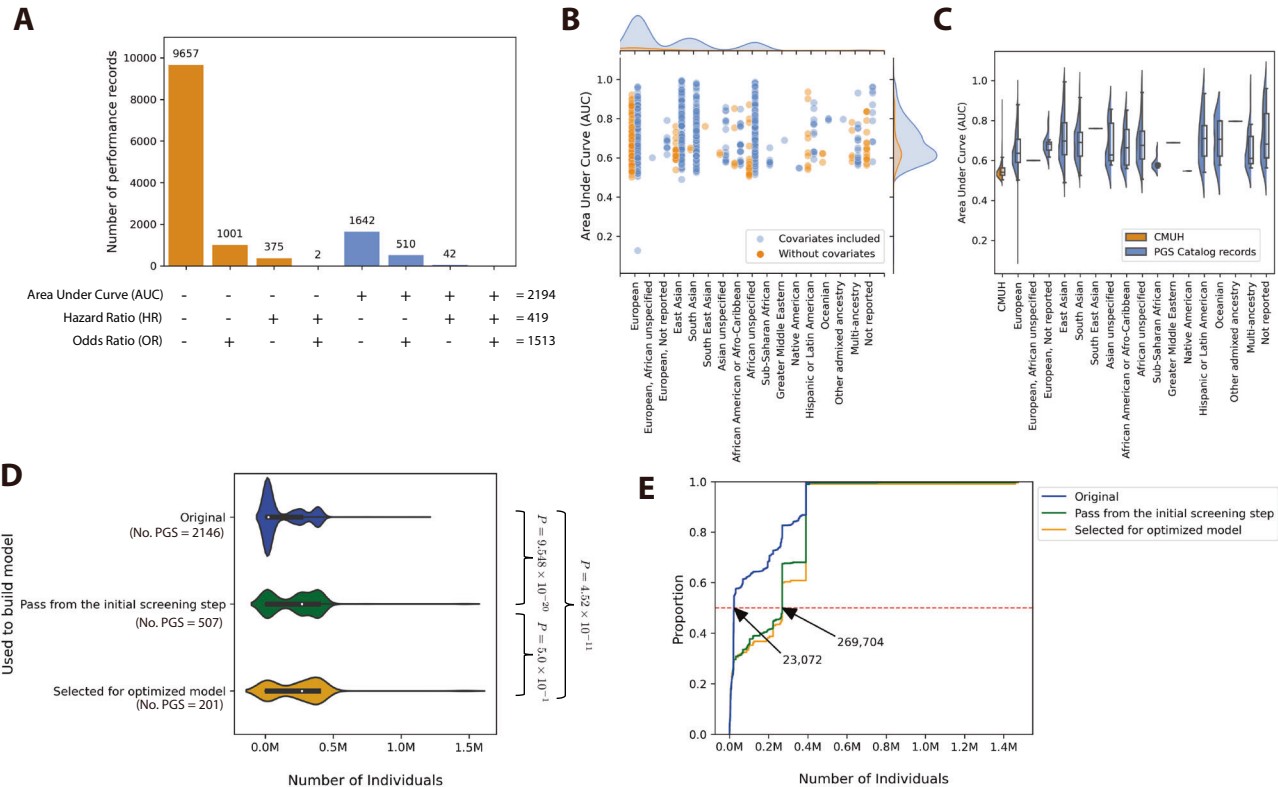

**Fig. 1 | Distribution of performance measurements and the number of individuals in the Polygenic Score Catalog. A** Distribution of performance measurement records. **B** Distribution of AUC values and covariate usage across different ancestry cohorts. In **A** and **B** The orange plot represents records in the PGS catalog that do not considerate of covariates; the blue plot represents records that considerate of covariates. **C** Comparison of the distribution of AUC values between AUC and different ancestry cohorts. The orange box represents the AUC values in the CMUH model; the blue band represents the AUC values recorded from the PGS catalog. **D** Distribution of the number of individuals at different process stages. The blue plot represents the PGS record used before the initial screening step. The green plot represents the PGS record used after the initial screening step. The orange plot represents the PGS record used for optimized model. In **C** and **D**, the

box represents the interquartile range (IQR), which spans from the 25th percentile (Q1) to the 75th percentile (Q3) of the data. The bottom and top edges of the box represent the smallest observation and the largest observation excluding outliers. The line inside the box represents the median (50th percentile) of the data. As for the violin plot, a smoothed kernel density estimate of the data distribution within each group is displayed. The bottom and top edges display the minimum and maximum values of the data. The two-sided Wilcoxon rank-sum test was used to calculate the $P$ value. Bold text indicates that the $P$ value < $1 \times 10^{-5}$ **E** Cumulative distribution of the number of individuals at each process stage. The blue line represents the PGS record used before the initial screening step. The green line represents the PGS record used after the initial screening step. The orange line represents the PGS record used for optimized model.

to retain PGS with relatively larger sample sizes. Similar results were observed in the final 201 PGS used for optimized models.

In our study, we consistently employed the AUC as the primary evaluation metric, utilizing four covariate inclusion strategies during model training: age and sex, PGS alone, PGS combined with sex and age, and PGS combined with sex, age, and the first four principal components. Evaluating the outcomes for 457 phenotypes based on the AUC achieved by the PGS model (Fig. 2A), we found that the majority of models exhibited enhanced AUC values with the addition of covariates. Setting a threshold of AUC > 0.6 to indicate effective model performance, we observed an increase in the number of phenotypes surpassing this threshold as more covariates were incorporated. Specifically, 24 phenotypes achieved an AUC > 0.6 for models trained with age and sex, 26 phenotypes for PGS alone, and 47 phenotypes for both models trained with PGS combined with age and sex and PGS combined with age, sex, and the first four principal components.

The distribution of data based on AUC values in the PGS catalog is illustrated in Fig. 2B. Examining the ancestry cohorts, we found that 58.352% of the data originated from calculations conducted on individuals of European descent, with East Asia and South Asia contributing 11.616% and 11.481% of the data, respectively. Among the 2194 records with AUC values, 1927 had covariates included in the calculations, while covariates were not considered in 267 records. Notably, regardless of the use of covariates, the disease identification

effectiveness of the PGS model predominantly fell within the AUC range of 0.6–0.7 in the PGS catalog, whereas our models fell within the AUC range of 0.5–0.6 (Table 1, Fig. 1C).

We explored whether changes in AUC across the three strategies for covariate inclusion are limited to specific disease classifications (Fig. 2B). The model trained with PGS, age, sex, and the first four principal components exhibited the highest performance, encompassing 213 out of 457 (46.61%) phenotypes. Following this, the model trained with PGS alone covered 157 out of 457 (34.35%) phenotypes, while the model trained with PGS, age, and sex covered 48 out of 457 (10.50%) phenotypes, and the model trained with age and sex covered 39 out of 457 (8.53%) phenotypes. This observed trend persisted across various disease classifications.

### Correlations between sample prevalence rates and other factors in the analysis

Our understanding of the allelic architecture in complex human diseases pertains to the patterns of genetic variations and their roles in influencing the risk or susceptibility of developing specific complex diseases. Studies have shown that low-frequency variants tend to exhibit greater penetrance in rare diseases, while common variants often display lower penetrance and require the presence of multiple variants, gene-gene interactions, or environmental factors to manifest as disease. Consequently, we hypothesized that as the prevalence rate

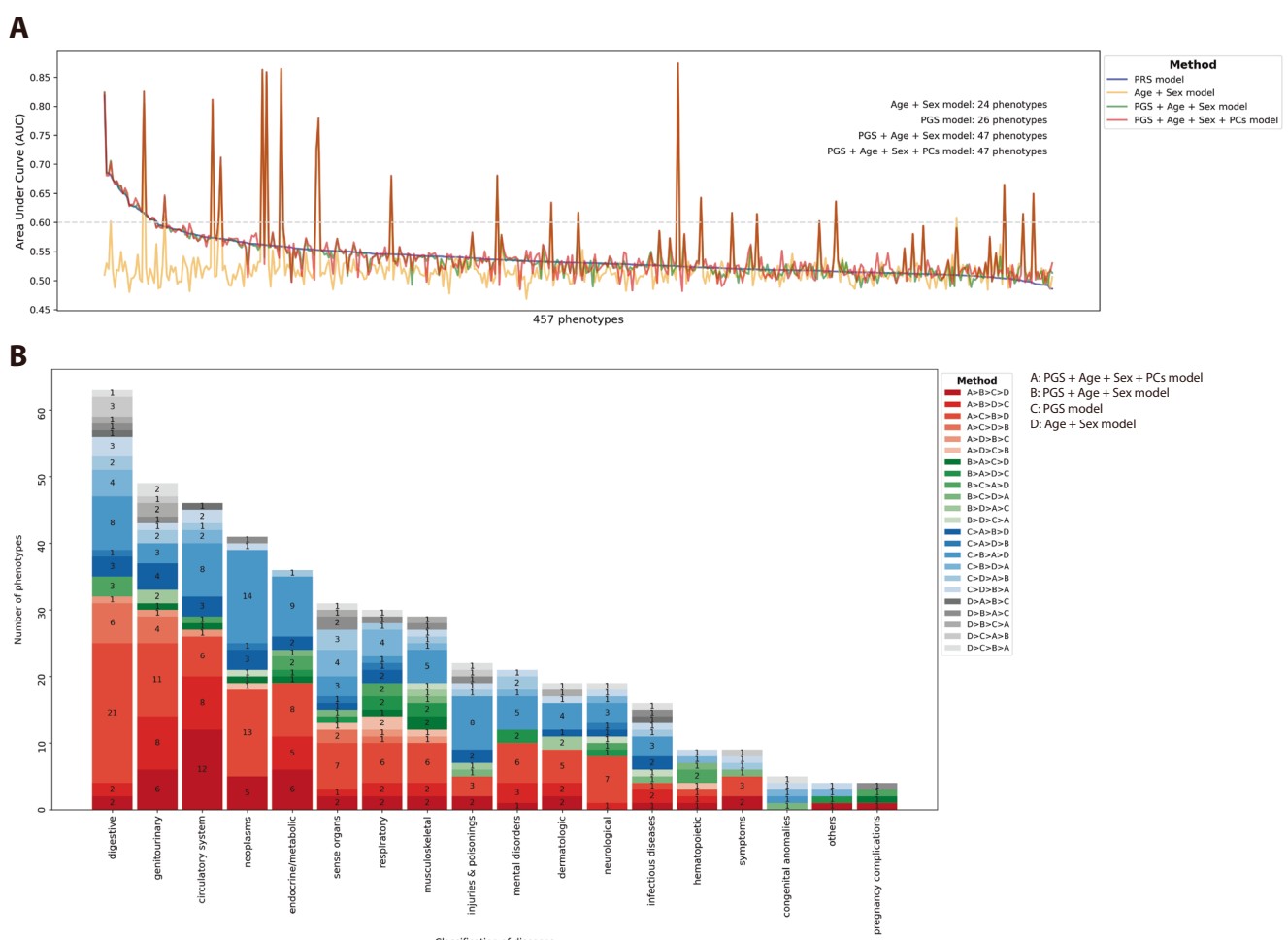

**Fig. 2 | Comparison of model performance with different covariate inclusion strategies. A** Changes in AUC across 457 phenotypes, sorted by AUC achieved by PGS models (The light gray dotted line represents AUC = 0.6). **B** Number of phenotypes exhibiting the AUC trend for the four covariate inclusion strategies. The red series represents the model trained with PGS, sex, age, and the first four principal components, which performed the best. The green series represents the model trained with PGS, sex, and age, which performed the best. The blue series represents the model trained with PGS alone, which performed the best. The gray series represents the model trained with sex and age, which performed the best.

## Table 1 | AUC distribution in terms of covariate usage

| AUC ranges | PGS catalog | | | PGS calculation in the CMUH | | | |
| --- | --- | --- | --- | --- | --- | --- | --- |
| | Covariates included (N = 1927) | Without covariates (N = 267) | Total (N = 2194) | PGS + Age + Sex + PCs model (N = 457) | PGS + Age + Sex model (N = 457) | PGS model (N = 457) | Age + Sex model (N = 457) |
| (0.0, 0.1] | - | - | - | - | - | - | - |
| (0.1, 0.2] | 0.05% | - | 0.05% | - | - | - | - |
| (0.2, 0.3] | - | - | - | - | - | - | - |
| (0.3, 0.4] | - | - | - | - | - | - | - |
| (0.4, 0.5] | 0.05% | - | 0.05% | 1.97% | 5.25% | 3.28% | 16.63% |
| (0.5, 0.6] | 24.81% | 28.52% | 25.26% | **84.25%** | **83.37%** | **90.15%** | **78.12%** |
| (0.6, 0.7] | **39.11%** | **50.00%** | **40.43%** | 10.94% | 8.53% | 5.91% | 3.50% |
| (0.7, 0.8] | 24.30% | 11.11% | 22.69% | 1.31% | 1.31% | 0.44% | 0.44% |
| (0.8, 0.9] | 8.92% | 8.89% | 8.91% | 1.53% | 1.53% | 0.22% | 1.31% |
| (0.9, 1.0] | 2.77% | 1.48% | 2.61% | - | - | - | - |

Bold text indicates the group with the largest proportion.

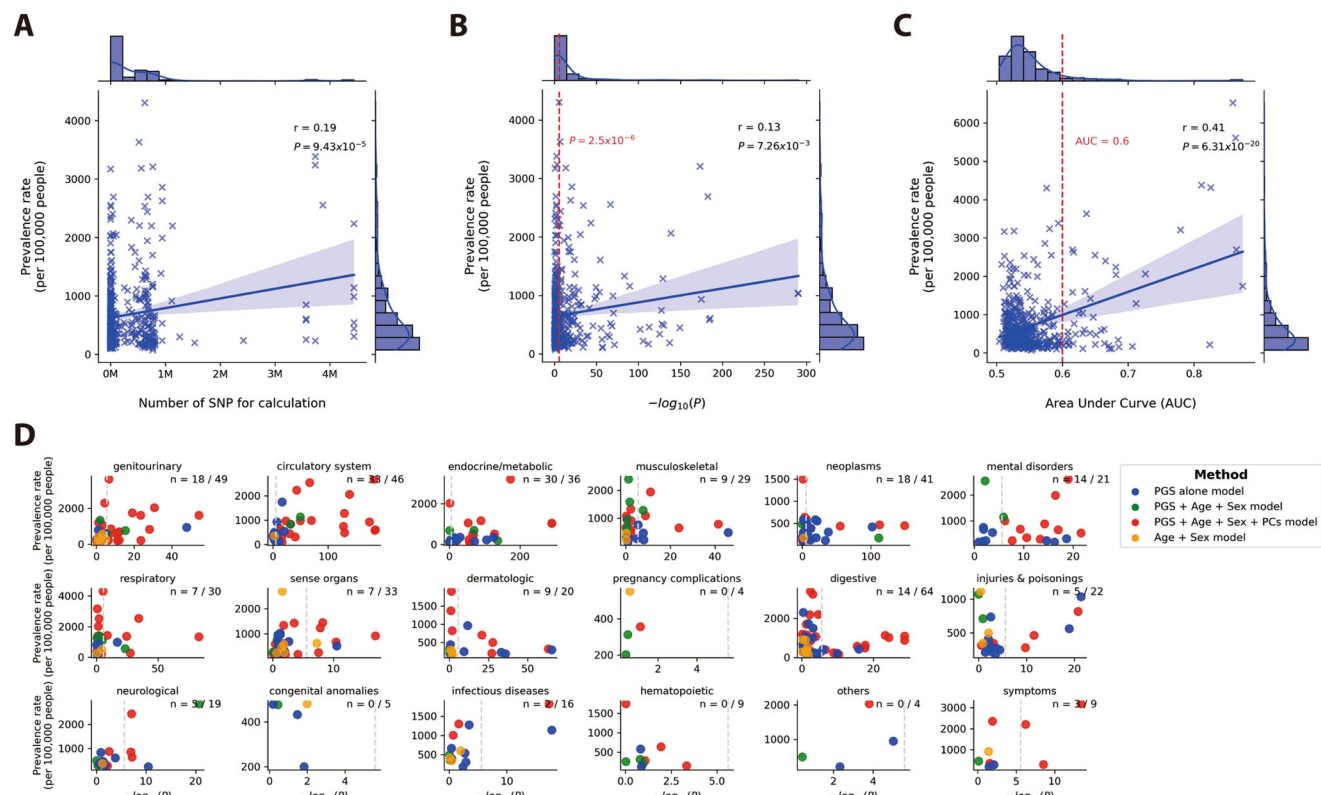

**Fig. 3 | Sample prevalence of the disease in CMUH correlation comparison.**
**A** Association between sample prevalence rate and number of SNPs used for PGS calculations. **B**) Association between sample prevalence rate and P values obtained from the Wilcoxon rank sum test of PGS distributions between case and control populations. (The red dotted line represents P values = $2.5 \times 10^{-6}$) **C** Association between the sample prevalence rate and AUC values for 457 phenotypes. (The red dotted line represents AUC = 0.6) In **A**–**C**, A linear regression line was plotted and

the confidence interval around the regression line was set to 95%. Pearson correlation coefficient (r) is a measure of the strength and direction of the linear relationship between two variables, ranging from −1 to 1. The P value is the probability of obtaining the observed correlation coefficient with the confidence interval is set to 95%. **D** Classification of diseases (n = phenotypes counted with P values less than $2.5 \times 10^{-6}$/total phenotypes; The light gray dotted line represents P values = $2.5 \times 10^{-6}$).

of a disease in a sample increases, the complexity of predictive models for that disease is likely to increase as well, along with the need for optimal covariance in disease prediction models.

Based on our dataset, there is a significant association between the sample prevalence rate and the number of SNPs used for PGS calculations, with a Pearson correlation coefficient of 0.19 and a P value of $9.43 \times 10^{-5}$ (Fig. 3A). Similarly, a significant association was observed between the sample prevalence rate and the -log10(p) obtained from

the Wilcoxon rank sum test of PGS distributions between case and control populations, with a Pearson correlation coefficient of 0.13 and a P value of $7.26 \times 10^{-3}$ (Fig. 3B). Furthermore, AUC values for the 457 phenotypes showed a more substantial association, with a Pearson correlation coefficient of 0.41 and a P value of $6.31 \times 10^{-20}$ (Fig. 3C).

Although the p-values obtained from the Wilcoxon rank sum test $<2.5 \times 10^{-6}$ mainly fell within circulatory system diseases (n = 33/46) and endocrine/metabolic diseases (n = 30/36; Fig. 3D), a distinct

pattern was still evident. Diseases with lower prevalence tend to have a lower -log10(p) value from Wilcoxon rank-sum test results, with optimal models relying largely on PGS alone. Conversely, diseases with higher prevalence typically yield a higher -log10(p) value from Wilcoxon rank sum results, necessitating additional covariates for the development of an optimal model.

## Performance of the PGS model in the CMUH dataset

Among the 457 phenotypes analyzed, 192 exhibited significant differences in distribution, with $P$ values obtained from the Wilcoxon rank sum test that were less than $2.5 \times 10^{-6}$. We identified a notable positive correlation between the AUC values and the $-log10(p)$ values obtained from the Wilcoxon rank sum test, with a Pearson correlation coefficient of 0.65 and a $P$ value of $1.06 \times 10^{-55}$ (Fig. 4A). Although the majority of AUC values fell between 0.5 and 0.6, four phenotypes achieved AUC values between 0.7 and 0.8, and seven phenotypes achieved AUC values between 0.8 and 0.9 (Fig. 4B). The lowest predictive performance was observed for oral aphthae, with an AUC of 0.504 (Fig. 4C), while the highest predictive performance was recorded for hyperplasia of the prostate, with an AUC of 0.874 (Fig. 4D).

Upon examining the distribution of PGSs in individuals affected by the disease, we observed a normal distribution in hyperplasia of the prostate but a skewed distribution in oral aphthae, characterized by sudden increases or decreases. We further investigated the relationship between PGS percentiles and patient prevalence, calculated using 100 equally sized quantiles in PGSs. For hyperplasia of the prostate, the mean patient prevalence at each percentile increased with rising PGS

percentiles, indicating an S-shaped curve and suggesting a strong nonlinear relationship between PGS percentiles and patient prevalence. Higher PGS quantiles corresponded to significantly elevated disease risk, highlighting the utility of PGS in identifying individuals at heightened risk of hyperplasia of the prostate. Conversely, plots for oral aphthae did not reveal a discernible relationship between PGS percentiles and patient prevalence at each percentile.

## Comprehensive comparison with other evaluation metrics

In addition to assessing PGSs using the Wilcoxon rank sum test and AUC, we calculated sensitivity, specificity, accuracy, precision, and recall to provide a more comprehensive evaluation of the model's performance. Among the 49 phenotypes identified with an AUC > 0.6 in the logistic regression model, six phenotypes did not meet the threshold of $2.5 \times 10^{-6}$ in the Wilcoxon rank sum test (Table 2), highlighting exceptions between statistical and practical significance. The lower $P$ value in the Wilcoxon rank sum test did not necessarily translate to superior model outcomes in the logistic regression model, and vice versa. Across the phenotypes, the logistic regression model exhibited high accuracy but low precision. Sensitivity, specificity, and recall showed varying results with no clear trends.

During the comparison of model validation results across different racial populations from the PGS Catalog records, we identified six diabetes-related phenotypes in our dataset that utilized the same PGS score file. Notably, the predictive performance for type 2 diabetes achieved an impressive AUC of 0.825, surpassing results in the PGS Catalog records for the East Asian dataset, which had an AUC of 0.810.

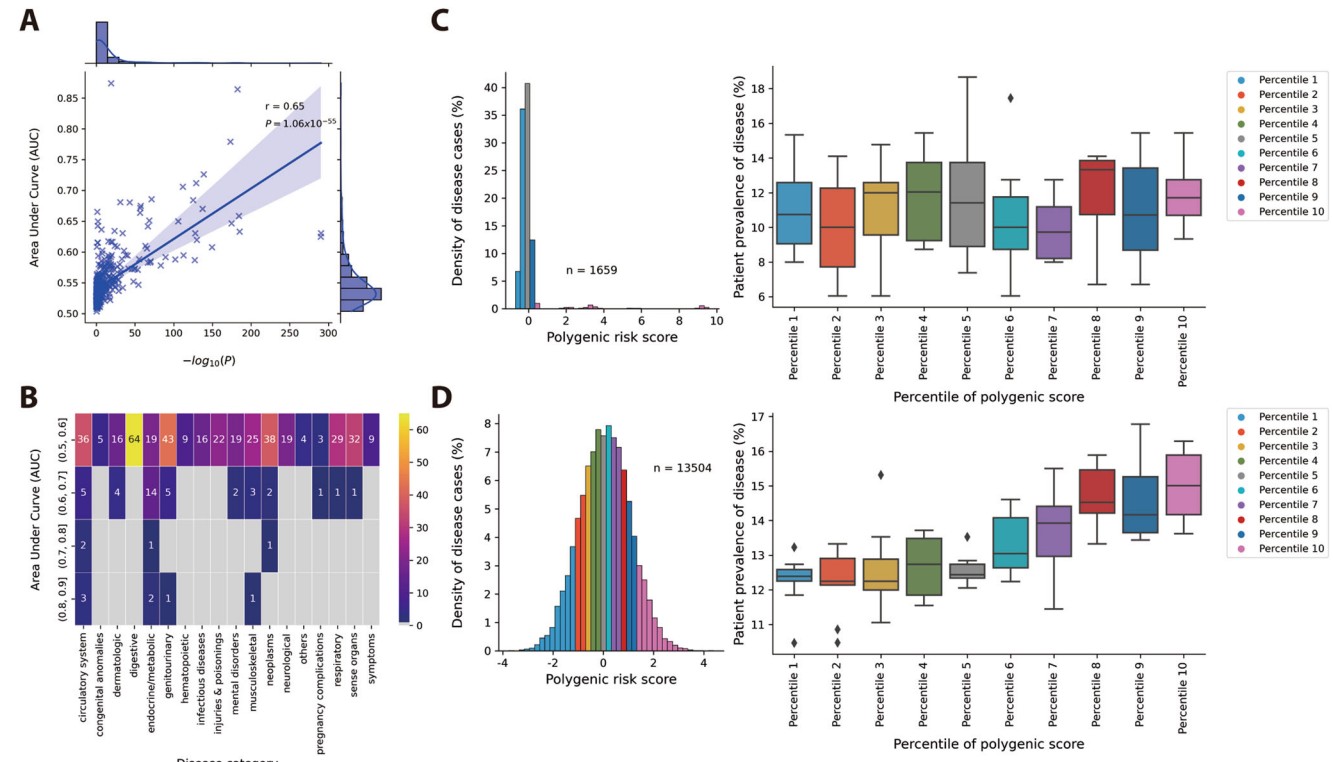

**Fig. 4 | Differential relationship between PGS performance and disease prevalence. A** Association between $P$ values obtained from the Wilcoxon rank sum test and AUC values of 457 phenotype−PGS pairs of traits. A linear regression line was plotted and the confidence interval around the regression line was set to 95%. Pearson correlation coefficient (r) is a measure of the strength and direction of the linear relationship between two variables, ranging from −1 to 1. The $P$ value is the probability of obtaining the observed correlation coefficient with the confidence interval is set to 95%. **B** AUC distribution of disease categories. **C** PGS distribution of patients with oral aphthae (n = patient number) and the relationship between PGS

percentiles and patient prevalence. **D** PGS distribution of patients with prostate hyperplasia (n = patient number) and the relationship between PGS percentiles and patient prevalence. In **C** and **D**, the box represents the interquartile range (IQR), which spans from the 25th percentile (Q1) to the 75th percentile (Q3) of the data. The bottom and top edges of the box represent the smallest observation and the largest observation excluding outliers. The line inside the box represents the median (50th percentile) of the data. Observations outside this range are considered outliers and are plotted individually.

**Table 2 | Results of evaluation metrics for 49 phenotypes**

| Category and phenotypes | PRS performance in CMUH | | | | | | | | | PRS performance from original data (AUC) | | |
|---|---|---|---|---|---|---|---|---|---|---|---|---|
| | PGSID | P-value | Model | AUC | Sensitivity | Specificity | Accuracy | Precision | Recall | Asian | European | African American |
| **sense organs** | | | | | | | | | | | | |
| Dizziness and giddiness (Light-headedness and vertigo) | PGS001959 | $2.61 \times 10^{-02}$ | Age + Sex model | 0.615 | 0.511 | 0.697 | 0.676 | 0.178 | 0.511 | - | - | - |
| **respiratory** | | | | | | | | | | | | |
| Acute pharyngitis | PGS001308 | $1.19 \times 10^{-02}$ | PGS + Age + Sex model | 0.643 | 0.648 | 0.674 | 0.671 | 0.200 | 0.648 | 0.758 | 0.772 | - |
| **pregnancy complications** | | | | | | | | | | | | |
| Other perinatal conditions of fetus or newborn | PGS001852 | $2.93 \times 10^{-01}$ | Age + Sex model | 0.608 | 0.504 | 0.711 | 0.691 | 0.158 | 0.504 | - | 0.571 | - |
| **neoplasms** | | | | | | | | | | | | |
| Cancer of prostate | PGS002240 | $1.99 \times 10^{-112}$ | PGS + Age + Sex model | 0.706 | 0.592 | 0.726 | 0.712 | 0.212 | 0.592 | - | - | - |
| Thyroid cancer | PGS001799 | $1.68 \times 10^{-34}$ | PGS model | 0.631 | 0.646 | 0.569 | 0.578 | 0.162 | 0.646 | - | 0.676 | - |
| Breast cancer [female] | PGS000335 | $4.61 \times 10^{-150}$ | PGS + Age + Sex + PCs model | 0.608 | 0.545 | 0.611 | 0.604 | 0.145 | 0.545 | - | - | - |
| **musculoskeletal** | | | | | | | | | | | | |
| Ankylosing spondylitis | PGS002089 | $<5 \times 10^{-324}$ | PGS + Age + Sex model | 0.824 | 0.669 | 0.943 | 0.912 | 0.601 | 0.669 | - | - | - |
| Osteoarthrosis | PGS001996 | $5.89 \times 10^{-02}$ | PGS + Age + Sex model | 0.665 | 0.593 | 0.685 | 0.675 | 0.190 | 0.593 | - | - | - |
| Osteoarthritis; localized | PGS001536 | $1.36 \times 10^{-02}$ | PGS + Age + Sex model | 0.615 | 0.466 | 0.760 | 0.727 | 0.198 | 0.466 | 0.828 | 0.699 | - |
| Osteoarthrosis, localized, primary | PGS002094 | $1.64 \times 10^{-08}$ | PGS + Age + Sex model | 0.603 | 0.430 | 0.772 | 0.734 | 0.191 | 0.430 | - | - | - |
| **mental disorders** | | | | | | | | | | | | |
| Other mental disorder | PGS000869 | $6.62 \times 10^{-03}$ | PGS + Age + Sex model | 0.649 | 0.789 | 0.457 | 0.504 | 0.192 | 0.789 | - | - | - |
| Anxiety disorders | PGS000907 | $9.78 \times 10^{-20}$ | PGS + Age + Sex + PCs model | 0.617 | 0.584 | 0.625 | 0.620 | 0.159 | 0.584 | - | - | - |
| **genitourinary** | | | | | | | | | | | | |
| Hyperplasia of prostate | PGS002076 | $1.01 \times 10^{-19}$ | PGS + Age + Sex + PCs model | 0.874 | 0.862 | 0.743 | 0.759 | 0.339 | 0.862 | - | - | - |
| Renal failure | PGS000708 | $1.72 \times 10^{-31}$ | PGS + Age + Sex + PCs model | 0.681 | 0.596 | 0.708 | 0.695 | 0.207 | 0.596 | - | 0.561 | - |
| Urinary tract infection | PGS002075 | $2.35 \times 10^{-07}$ | PGS + Age + Sex + PCs model | 0.636 | 0.550 | 0.693 | 0.677 | 0.180 | 0.550 | - | - | - |
| Chronic renal failure [CKD] | PGS000884 | $3.58 \times 10^{-24}$ | PGS + Age + Sex + PCs model | 0.634 | 0.491 | 0.760 | 0.730 | 0.206 | 0.491 | - | - | - |
| Abnormal results of function study of kidney | PGS002237 | $4.76 \times 10^{-16}$ | PGS + Age + Sex model | 0.618 | 0.479 | 0.746 | 0.716 | 0.194 | 0.479 | - | 0.75 | - |
| Elevated prostate specific antigen [PSA] | PGS002240 | $2.35 \times 10^{-24}$ | PGS + Age + Sex + PCs model | 0.610 | 0.410 | 0.760 | 0.723 | 0.169 | 0.410 | - | - | - |
| **endocrine/metabolic** | | | | | | | | | | | | |
| Type 2 diabetes | PGS002308 | $<5 \times 10^{-324}$ | PGS + Age + Sex + PCs model | 0.825 | 0.822 | 0.689 | 0.707 | 0.293 | 0.822 | 0.81 | 0.793 | 0.848 |
| Diabetes mellitus | PGS002354 | $<5 \times 10^{-324}$ | PGS + Age + Sex + PCs model | 0.811 | 0.828 | 0.677 | 0.698 | 0.292 | 0.828 | - | - | - |
| Disorders of lipoid metabolism | PGS002334 | $7.76 \times 10^{-174}$ | PGS + Age + Sex + PCs model | 0.779 | 0.847 | 0.611 | 0.642 | 0.247 | 0.847 | - | - | - |
| Polyneuropathy in diabetes | PGS002308 | $6.17 \times 10^{-72}$ | PGS model | 0.686 | 0.788 | 0.495 | 0.530 | 0.172 | 0.788 | 0.81 | 0.793 | 0.848 |
| Diabetic retinopathy | PGS002308 | $9.16 \times 10^{-126}$ | PGS model | 0.684 | 0.795 | 0.491 | 0.527 | 0.171 | 0.795 | 0.81 | 0.793 | 0.848 |
| Diabetes type 2 with peripheral circulatory disorders | PGS002379 | $4.08 \times 10^{-58}$ | PGS + Age + Sex model | 0.675 | 0.530 | 0.747 | 0.722 | 0.215 | 0.530 | - | - | - |
| Type 1 diabetes | PGS001371 | $1.71 \times 10^{-57}$ | PGS + Age + Sex + PCs model | 0.671 | 0.435 | 0.853 | 0.810 | 0.251 | 0.435 | - | - | - |
| Type 2 diabetes with ophthalmic manifestations | PGS002308 | $7.32 \times 10^{-138}$ | PGS + Age + Sex model | 0.670 | 0.555 | 0.713 | 0.697 | 0.184 | 0.555 | 0.81 | 0.793 | 0.848 |
| Morbid obesity | PGS002679 | $4.20 \times 10^{-57}$ | PGS + Age + Sex + PCs model | 0.664 | 0.671 | 0.617 | 0.624 | 0.192 | 0.671 | - | - | - |
| Type 2 diabetes with renal manifestations | PGS002308 | $<5 \times 10^{-324}$ | PGS + Age + Sex model | 0.664 | 0.676 | 0.573 | 0.584 | 0.166 | 0.676 | 0.81 | 0.793 | 0.848 |
| Obesity | PGS002679 | $3.69 \times 10^{-69}$ | PGS + Age + Sex + PCs model | 0.651 | 0.697 | 0.541 | 0.558 | 0.156 | 0.697 | - | - | - |
| Type 2 diabetes with neurological manifestations | PGS002308 | $2.20 \times 10^{-106}$ | PGS model | 0.650 | 0.574 | 0.649 | 0.640 | 0.179 | 0.574 | 0.81 | 0.793 | 0.848 |

**Table 2 (continued) | Results of evaluation metrics for 49 phenotypes**

| Phenotype | PGS (CMUH) | P value in CMUH | Model | | | | | | | PRS performance from original data (AUC) | |
|---|---|---|---|---|---|---|---|---|---|---|---|
| Overweight, obesity and other hyperalimentation | PGS001943 | **2.84 × 10⁻⁶⁷** | PGS + Age + Sex + PCs model | 0.642 | 0.654 | 0.590 | 0.597 | 0.162 | 0.654 | - | - |
| Gout and other crystal arthropathies | PGS001789 | **7.18 × 10⁻²⁹¹** | PGS + Age + Sex + PCs model | 0.631 | 0.565 | 0.636 | 0.628 | 0.166 | 0.565 | - | 0.807 |
| Gouty arthropathy | PGS001789 | **2.15 × 10⁻¹²⁸** | PGS + Age + Sex + PCs model | 0.631 | 0.534 | 0.676 | 0.660 | 0.173 | 0.534 | - | 0.807 |
| Gout | PGS001789 | **1.21 × 10⁻²⁹⁰** | PGS + Age + Sex + PCs model | 0.625 | 0.555 | 0.639 | 0.630 | 0.165 | 0.555 | - | 0.807 |
| Thyroiditis | PGS001181 | **1.58 × 10⁻²¹** | PGS + Age + Sex + PCs model | 0.616 | 0.622 | 0.556 | 0.564 | 0.152 | 0.622 | 0.715 | 0.767 |
| **dermatologic** | | | | | | | | | | | |
| Lupus (localized and systemic) | PGS000328 | **6.95 × 10⁻⁶⁴** | PGS + Age + Sex + PCs model | 0.610 | 0.515 | 0.651 | 0.637 | 0.147 | 0.515 | - | 0.83 |
| Psoriasis | PGS002083 | **9.77 × 10⁻²⁸** | PGS + Age + Sex + PCs model | 0.609 | 0.619 | 0.555 | 0.562 | 0.151 | 0.619 | - | - |
| Systemic lupus erythematosus | PGS000328 | **4.07 × 10⁻⁶⁶** | PGS model | 0.606 | 0.496 | 0.681 | 0.662 | 0.154 | 0.496 | - | 0.83 |
| Psoriasis vulgaris | PGS002344 | **1.59 × 10⁻³⁶** | PGS model | 0.602 | 0.475 | 0.714 | 0.687 | 0.173 | 0.475 | - | - |
| **circulatory system** | | | | | | | | | | | |
| Hypertensive heart disease | PGS002701 | **3.34 × 10⁻¹⁸³** | PGS + Age + Sex + PCs model | 0.865 | 0.827 | 0.741 | 0.751 | 0.292 | 0.827 | - | - |
| Essential hypertension | PGS002701 | **<5 × 10⁻³²⁴** | PGS + Age + Sex + PCs model | 0.863 | 0.820 | 0.757 | 0.769 | 0.454 | 0.820 | - | - |
| Hypertension | PGS002701 | **<5 × 10⁻³²⁴** | PGS + Age + Sex + PCs model | 0.859 | 0.847 | 0.720 | 0.748 | 0.464 | 0.847 | - | - |
| Ischemic Heart Disease | PGS000337 | **2.42 × 10⁻¹³⁹** | PGS + Age + Sex + PCs model | 0.726 | 0.755 | 0.595 | 0.613 | 0.188 | 0.755 | 0.674 | - |
| Other hypertensive complications | PGS002047 | **9.52 × 10⁻¹³⁰** | PGS + Age + Sex + PCs model | 0.712 | 0.647 | 0.692 | 0.687 | 0.207 | 0.647 | - | - |
| Cerebrovascular disease | PGS002725 | **9.56 × 10⁻⁴⁴** | PGS + Age + Sex + PCs model | 0.681 | 0.631 | 0.676 | 0.671 | 0.198 | 0.631 | 0.765 | - |
| Atrial fibrillation | PGS000331 | **4.95 × 10⁻¹⁸⁵** | PGS + Age + Sex + PCs model | 0.659 | 0.697 | 0.550 | 0.566 | 0.159 | 0.697 | - | - |
| Coronary atherosclerosis | PGS000337 | **2.21 × 10⁻¹⁷⁵** | PGS + Age + Sex + PCs model | 0.647 | 0.613 | 0.606 | 0.607 | 0.163 | 0.613 | 0.674 | - |
| Atrial fibrillation and flutter | PGS000331 | **6.40 × 10⁻¹⁸⁶** | PGS + Age + Sex + PCs model | 0.633 | 0.599 | 0.611 | 0.610 | 0.155 | 0.599 | - | - |
| Myocardial infarction | PGS002361 | **1.49 × 10⁻¹³⁰** | PGS + Age + Sex + PCs model | 0.620 | 0.681 | 0.509 | 0.529 | 0.154 | 0.681 | 0.674 | - |

Bold text indicates that the P value < $2.5 \times 10^{-6}$.

However, the predictive power of other complications stemming from type 2 diabetes ranged from 0.65 to 0.75, indicating the need for additional genetic factors or information such as age of onset, duration time, and degree of control of type 2 diabetes to enhance predictive accuracy further.

Furthermore, regarding type 1 diabetes, our model achieved an AUC of 0.671. Interestingly, comparison with results from the PGS catalog records revealed that the model in the catalog, trained with European data, performed better on the East Asian population (AUC = 0.893) than on the European population (AUC = 0.705) in the testing dataset. Upon closer examination, we noted a significant difference in the dataset sizes used for these assessments. The dataset used for calculating AUC in the East Asian population included 5 cases and 1699 controls, while the European dataset comprised 186 cases and 24,719 controls. This discrepancy indicates that the assessments were strongly influenced by randomness or sampling bias.

## Discussion

In recent years, PGSs have gained popularity in genomic research, with researchers typically allocating 60%–80% of a dataset for GWAS and the remaining 20%–40% for PGS calculations to develop personalized risk models[8,9]. However, evaluating the effects of PGS in independent datasets has been challenging due to the necessity of a sufficiently large sample size[22]. Fortunately, the introduction of the PGS Catalog[21] in 2021 has addressed this issue, allowing for exploration of the clinical significance of PGS.

The PGS Catalog team curates PGSs from published studies using standardized formats and ontologies to ensure the consistency and comparability of PGS data. This enables researchers to compare different PGSs for the same trait, evaluate overall predictive performance, and assess applicability in new populations and contexts. The first study to construct PGSs using weights curated from the PGS Catalog was published in Epidemiology and Global Health Genetics and Genomics in March 2023[23]. This study focused on PGSs for breast, prostate, colorectal, and lung cancers in 21,694 East Asian individuals. While the PGSs demonstrated predictive power, with AUCs ranging from 0.58 to 0.70, the study indicated that appropriate correction factors may be necessary to improve calibration.

To explore whether the predictive power of PGSs extends beyond cancer, we systematically calculated PGSs across 457 phenotypes using score files from the PGS Catalog. Our findings revealed a positive correlation between the ability of PGSs to predict disease risk and the prevalence rate of the relevant disease in a population. This correlation may stem from larger sample sizes, which enhance statistical power to unveil associations between genetic variants and diseases, thereby constructing more reliable PGSs[24]. Additionally, diseases with a high prevalence rate may exhibit a genetic architecture influenced by specific traits or factors that affect the development and efficacy of PGSs[25]. Certain polygenic structures may be associated with multiple generalized traits or diseases.

When comparing two common metrics for evaluating PGS, namely the $P$ value obtained from a Wilcoxon rank sum test and the AUC of a logistic regression model, a strong correlation was observed between the two. However, differences were noted for certain phenotypes, likely attributed to underlying test assumptions, effect sizes, sample sizes, and methods of constructing PGSs or adjusting for covariates in the logistic regression model[26]. Currently, no established criteria exist for determining the suitability of a PGS model for clinical use. Therefore, multiple matrices should be considered to provide a reliable assessment of PGS performance.

Among the 457 analyzed phenotypes, AUC values ranged from 0.504 to 0.874, with 408 (89.28%) falling within the 0.5–0.6 range. These findings indicate challenges in using PGSs in other studies, primarily due to most PGSs in the PGS Catalog being derived from individuals of European descent, raising concerns about heterogeneity across different populations. Unlike genome-wide association study (GWAS) meta-analyses, where data harmonization and integration across diverse populations are common practices, the SNPs recorded in these PGS score files have undergone various filtering procedures. These processes include addressing factors such as population stratification, linkage disequilibrium trimming, aggregation of summary statistics, and applying significance thresholding. As a result, the data in these PGS score files are optimized for their original purposes but may pose challenges when attempting to incorporate new data or make further adjustments with different population groups. Furthermore, although the PGS Catalog includes models incorporating environmental factors or gene-environment interactions, users can only obtain PGS scoring formulas without the impact/weight related to environmental factors, potentially complicating model application and reducing AUC[27].

Despite these challenges, we identified 49 phenotypes with AUC values > 0.6, indicating that certain genetic variants have consistent effects on traits or diseases across different populations. Notable examples include type 2 diabetes, type 1 diabetes, pathological obesity, gout, chronic thyroiditis, myocardial infarction, atrial fibrillation, ankylosing spondylitis, prostate cancer, thyroid cancer, and breast cancer, among others (Table 2). Previous research from our team has demonstrated significant associations between genetic variants and conditions such as gout[28], hyperthyroidism[29], obesity[30], and various types of cancer[31]. These findings underscore the potential for further investigation to identify and characterize these genetic variants and explore their implications for disease risk. Addressing population-specificity issues may involve exploring additional integration or adjustment methods. Incorporating environmental factors and gene-environment interactions into PGS models could enhance the accuracy and robustness of risk predictions. Further research is essential to fully leverage the potential of PGSs in advancing precision medicine and improving public health outcomes.

## Methods
### Study population and genetic data quality control
From 2018 to 2021, a cohort of 347,954 patients was recruited from the outpatient department of CMUH with the approval of the Institutional Review Board (IRB number: CMUH107-REC3-058 (AR-1); date of approval: 07/20/2018). Age of participants was calculated as of December 31, 2021, using the formula: Age = December 31, 2021 minus date of birth. The sex of participants was determined based on the SNP array genotyping result. Genotyping was performed using the Axiom Genome-Wide 1.0 customized array plate (Affymetrix, Santa Clara, CA, USA), in accordance with their guidelines and regulations (Santa Clara, CA, USA). Preimputation quality control of the genotype data was conducted using PLINK 2.0 (https://www.cog-genomics.org/plink2)[32]. SNPs and individuals were excluded if the missing rate was >10%, the minor allele frequency was <0.01, the Hardy–Weinberg equilibrium exact test was less than $1 \times 10^{-6}$, or the total call rate was <0.98[33]. Phased genotype data were subsequently imputed using beagle 5.4 (version: 22 Jul 22.46e) (https://faculty.washington.edu/browning/beagle/beagle.html)[34] with a Taiwan population-specific reference panel containing 1495 whole-genome sequencing data. After quality control and imputation, a total of 276,712 individuals with 14,029,683 variants were included for analysis.

### Phenotype identification and PGS trait paring
Since 1980, CMUH has provided treatment to over 3 million patients. Patient data, encompassing demographic information and clinical details such as medical history, medication history, and diagnostic test results, is systematically recorded in an electronic health record system. To identify phenotypes for analysis, we sourced the International Classification of Diseases version 9th and 10th (ICD) codes from medical records, along with patient demographics like age, sex, and other identifying information (Fig. 5). Utilizing the createPhenotypes function in PheWAS (https://github.com/PheWAS/PheWAS)[35], we grouped ICD codes to identify 1836 phenotypes (Supplementary

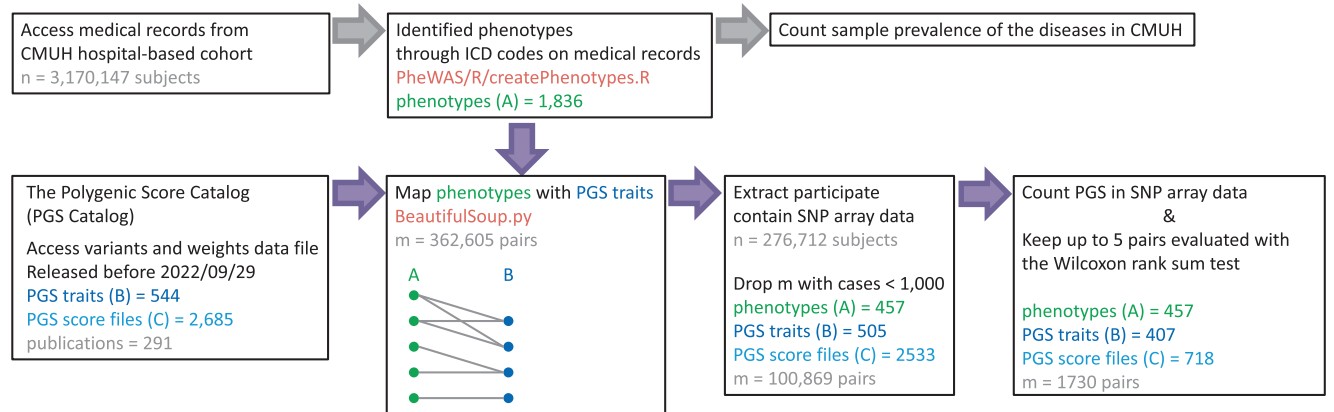

**Fig. 5 | Data collection and processing workflow for data from the China Medical University Hospital and the PGS Catalog.** In the figure, n is the number of subjects included in the analysis and m is the number of phenotype−PGS pairs.

Data 1). The inclusion and exclusion criteria were established using the Clinical Classification Software grouping schema and the incidence of codes in the electronic health records of several medical facilities, accessible at https://www.phewascatalog.org/phecodes.

We gathered variant and weight data files for 544 PGS traits from the PGS Catalog[21]. To link the phenotypes and PGS traits, we employed the surjective pairing method using the BeautifulSoup function (https://git.launchpad.net/beautifulsoup)[36], a widely-used Python library for web scraping. BeautifulSoup enabled us to search for and extract phenotype-related keywords from the HTML content of the PGS Catalog website, including specific HTML elements like tables or div tags. This effort resulted in a total of 362,605 phenotype−PGS pairs (Supplementary Data 2). However, only 100,869 of these phenotype −PGS pairs met the threshold for PGS performance analysis, based on the requirement of a sufficient sample size (>1000 cases) (Fig. 5). All the code used in this section is recorded in Supplementary Software 1.

We further scrutinized the retained phenotype-PGS pairs based on the *P*-value derived from the Wilcoxon rank sum test. This allowed for an initial screening of PGS traits, with up to 5 candidates considered per phenotype. This rigorous process led to the identification of 1730 phenotype-PGS pairs, which constituted the foundational dataset for our study.

### PGS model construction and predictive performance
The PGS Catalog offers an online user interface (https://www.pgscatalog.org/) and provides score files for various traits, featuring uniformly formatted columns for variations, alleles, and weights. The PGSs were computed using PRSice-2 (https://choishingwan.github.io/PRSice/)[37]. A weighted PGS model was employed, expressed as follows[38]:

$$PRS_w = \hat{\beta}_1 G_1 + \dots, \hat{\beta}_K G_K \qquad (1)$$

where $G_k (k = 1, \dots, K)$ represents the number of risk alleles for each genetic variant, which are coded as 0, 1, or 2 under the additive genetic model. The estimate of marginal genetic effects in the weighted SNP list is denoted by $\hat{\beta}_k (k = 1, \dots, K)$.

Following the computation, PGS distribution plots, stratified by disease case status (case-control groups), were generated using the ggplot2 R package. To assess the predictive capability of the PGS model, we implemented age- and sex-matching procedures at ratios of 1:8 for case-control pairs (refer to Supplementary Result, Supplementary Fig. 1), facilitated by the MatchIt package in R[39]. Subsequently, we conducted statistical tests to evaluate differences in PGS distributions between cases and controls. This involved performing a two-sided Welch's two-sample t-test and a two-sided Wilcoxon rank sum test. The dataset was then split into training and testing sets in an 8:2 ratio,

employing four different covariate inclusion strategies for training models: age and sex, PGS alone, PGS combined with sex and age, and PGS combined with sex, age, and the first four principal components. To assess the significance of the area under the curve (AUC), Delong's method[40] was used in conjunction with Youden's index (J) to determine the optimal J[41] cutoff point for the PGS. In the context of survival analysis, Cox proportional hazards models[42,43] were employed, utilizing age as the time scale to investigate the association between PGSs and disease endpoints[44]. Additionally, disease distribution plots were created, stratifying individuals based on their PGS percentiles. These plots were generated using the ggplot2 package in R (version 4.1.1) and compiled into the GeneAnaBase website (https://pgscatalog.azure.nihxcmuh.org/#/) for review (Fig. 6). All the code used in this section is recorded in Supplementary Software 1.

### Evaluate the consistency of PGSs in different SNP detection methods
During the calculation of PGS using SNP array data, we encountered a challenge as 14,029,683 variants did not meet the required criteria for the PGS score files. To ensure the integrity of PGS values, we conducted a comparative analysis involving 353 individuals who had both whole-genome sequencing and SNP array data.

In brief, the in-house whole-genome sequencing data was obtained using a 30X depth 150 bp paired-end sequencing method. This was followed by alignment to the GRCh38 human genome and completed variant calling using the DRAGEN genome pipeline (Illumina, Inc., San Diego, CA, USA). PGSs were calculated using PRSice-2, the same method used for the SNP array data. To assess the reliability of the PGS scoring from SNP array data, we utilized the intraclass correlation coefficient (ICC) approach, denoted by the formula[45]:

$$ICC = \frac{MSBS - MSE}{MSBS + (k-1)MSE} \qquad (2)$$

where MSBS represents the mean square between subjects, MSE represents the mean square error, and K represents the number of methods under consideration. This approach enables the evaluation of agreement between the two methods while treating them as fixed effects, thereby eliminating systematic errors and focusing on the random residual error. All the code used in this section is recorded in Supplementary Software 1.

The PGS score files exhibited a wide range of required SNPs, with an average of 28147.41 ± 73980.64 and a median of 420.00 variants. The average SNP missing rate in the SNP array data was 16.52% ± 14.05%, with a median of 11.28%. The average ICC score was computed at 0.82 ± 0.12, with a median value of 0.82. In our study, this ICC score

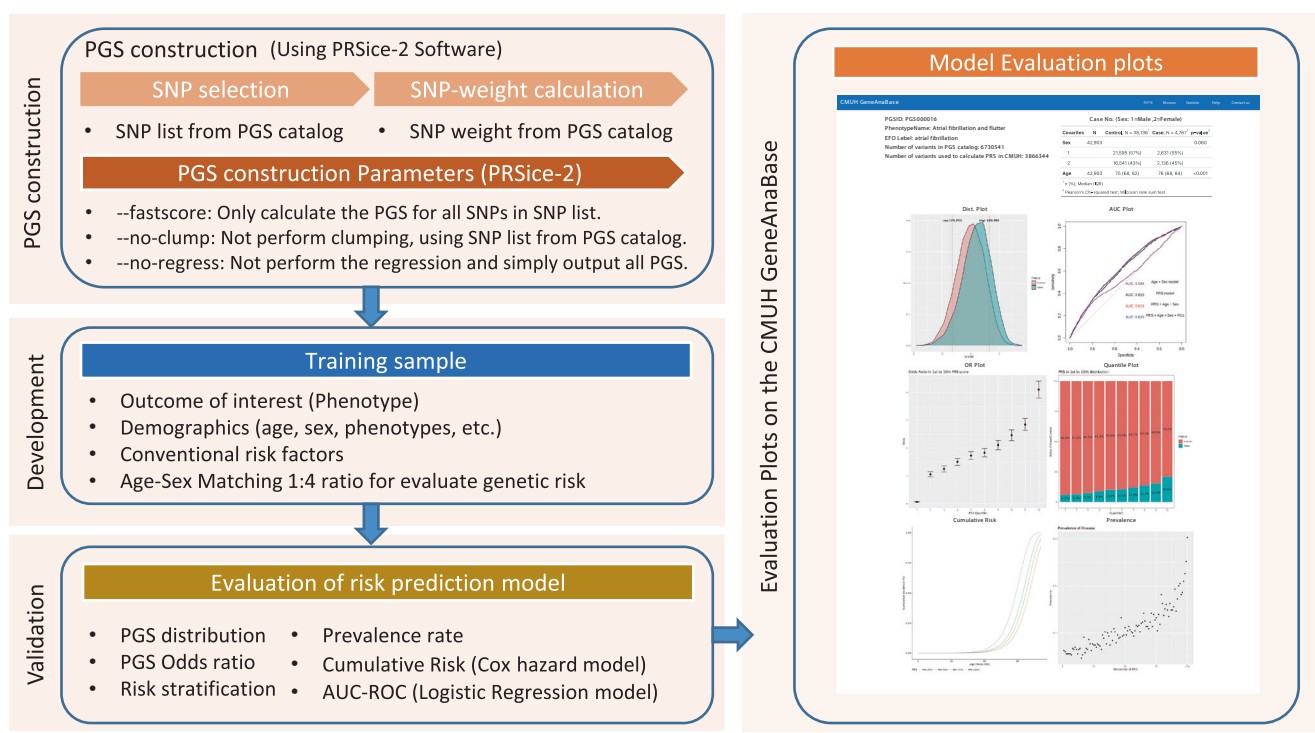

**Fig. 6 | Calculation, measurement, and display of the polygenic risk score model evaluation results.** The entire process can be divided into four parts, namely PGS construction, model development, model evaluation, and drawing evaluation diagrams on CMUH GeneAnaBase.

ranged between 0.75 and 1, indicating a high level of consistency across most comparisons.

## Reporting summary

Further information on research design is available in the Nature Portfolio Reporting Summary linked to this article.

## Data availability

The raw SNP array data are protected and are not available due to data privacy laws. However, we are committed to fostering collaboration and promoting transparency in scientific research. As such, we welcome collaborative projects and discussions with fellow researchers who may require access to the data. Please provide your identity, employer, purpose of data access, and IRB approval to the mailbox of the corresponding author, Dr. F.-J.T. (000704@tool.caaumed.org.tw). We will meet within one month to discuss and respond. The statistical results generated in this study are provided on the GeneAnaBase website, which can be found at https://pgscatalog.azure.nihxcmuh.org/#/. Source data are provided with this paper.

## Code availability

All codes used for data download, processing, calculation, and graphing are recorded in the Supplementary Software. Please refer to this document for detailed information.

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

## Acknowledgements

Thanks for allowing us to use the genotyping microarray data from the Million Person Precision Medicine Initiative in China Medical University Hospital. We are grateful to the Health Data Science Center, China Medical University Hospital, for providing administrative and technical support. This study was supported in part by the Taiwan Ministry of Health and Welfare Clinical Trial Center (MOHW112-TDU-B-212–144004), and was funded by China Medical University and China Medical University Hospital (DMR–112–124, DMR–111,227, MOST 110-2314-B-039-010-MY2, MOST 111-2321-B-039-005, MOST 111-2321-B-030-004, and MOST 111-2622-8-039-001-IE).

## Author contributions

F.-J.T. and K.-C.H. equally corresponded to the study's conception and provided essential resources. T.-H.S. originated and formulated the study design. C.-C.W. conducted data analysis. T.-H.S. and C.-C.W. authored the manuscript. T.-Y.L. and Y.-X.H. were involved in the data collection and preprocessing process. S.-C.L. and Y.-D.C. offered specialized knowledge in website and database development. S.-Y.C. facilitated research recommendations. All the authors reviewed and provided feedback for each draft of the manuscript, and all the authors have read and approved the final version of the manuscript.

## Competing interests

The authors declare no competing interests. All authors declare that they have no known competing financial interests or non-financial interests that could have appeared to influence the work reported in this paper.
