## [Peer Review File · Nature Communications]

Utility of Polygenic Scores Across Diverse Diseases in a Hospital Cohort for Predictive ModelingREVIEWER COMMENTS

Reviewer #1 (Remarks to the Author):

Sun et al. collated PRS weights (mainly from European ancestry) from PGS catalogue and performed a systematic calculation of PRSs using ad hoc six step steps across many traits (n=457) in a cohort of over a quarter of a million Taiwanese individuals from China Medical University Hospital to assess applicability of implementing risk prediction across the wide range of human traits. They used logistic regression models to calculate the adjusted area under the receiver operating characteristic curve (AUC) to evaluate the ability of PRSs to predict disease traits. This I feel is a great resource that will inform other researchers working in populations of East Asian ancestry on trait specific transferability of cross-ancestry disease predictions. However, the study has one major limitation/issues that need to be addressed for the reader to put these results in the contexts of all the factors that been previously established (by many other studies) to affect PRS prediction quality:

1. Several factors have previously been identified that will affect prediction quality of PRS – sample sizes of both source data and test data, genetic architecture of the trait, and of course the model used to generate the weights – all the factors they highlight in discussion while citing ref#39 – and I expected these factors to be assessed in making inferences above applicability and transferability of PRS predictions across ancestry in this manuscript. The authors
 - a. did not list and make determination on correlation of sample sizes of the source data (in which PGS weights) and the ones in their data with prediction quality – for example there observed lowest p value for T2D which has typically larger samples sized summary data, in addition to is the accuracy correlated with either PGS case samples sizes (including the size of the Asian samples in the GWAS) or size of the test samples.... They do however note that “....ability to predict disease risk was positively correlated with the prevalence of the relevant disease in a population.” And attribute it to sample sample sizes – a point they need to flesh out more clearly in the manuscript with appropriate statistics.
 - b. performed test in 1:4 case-control matched data but did not use the same criterion for filtering out PGS data downloaded from the catalogue – we have previously noted that the unbalanced summary data can affect prediction quality – Wang et al., 2022 - [https://www.cell.com/cell-genomics/pdf/S2666-979X\(22\)00204-X.pdf](https://www.cell.com/cell-genomics/pdf/S2666-979X(22)00204-X.pdf)
 - c. Besides noting that there is correlation of prediction quality with number of variants used, they did not establish what the overlap with their data was – this will affect the prediction quality.
 - d. The authors note that cross-ancestry “... disease prediction ability was highest for ankylosing spondylitis ...” but did not explore the reason which I expected them to do. This is in contrast to heart failure which we previously noted to have low heritability among 14 endpoints – pointing to the effect of genetic architecture on the prediction [https://www.cell.com/cell-genomics/pdf/S2666-979X\(22\)00204-X.pdf](https://www.cell.com/cell-genomics/pdf/S2666-979X(22)00204-X.pdf)

One minor point, the authors should change “features” to “disease” or “traits” on first line of the last paragraph in the last page of the discussion section,

Reviewer #2 (Remarks to the Author):

In this paper, Sun et al has performed standard polygenic risk scores analyses using effect sizes from the PGS catalog in 276k individuals from the China Medical University Hospital. While the analyses performed were reasonable, I do have a few questions regarding the presentation and interpretation of the results.

1. In the abstract, the authors stated that there were significant incremental predictive performance, however, in the method and result section, the authors do not seem to include any analyses that investigate the incremental performance of PGS, rather, only full model analyses (with covariates included) were included.
2. The PGS catalog provides a REST API that is relatively easy to use and will allow direct download of meta information in JSON format (<http://www.pgscatalog.org/rest/>). You may download the ancestry information, sample size and even the disease trait information directly without requiring scrapping, thus reduce chance of error.
3. In figure 1, it stated that extracting participants in the genetic project, was that intended to mean the sample size from CMUH? In addition, given there were only 544 PGS traits, how come there end up having more than 1,531 traits in the end?
4. Again, from figure 1, n and m were not annotated. I guess that'd be the sample size and number of traits. To improve the clarity, it might help to add a box between "Access medical records" and "Identified phenotype", stating the number of post QC genetic samples (276,712).
5. For the t test and wilcox test, were there any attempt to control for covariates such as age, sex, and population stratification? Was the metric reported based on the PGS only, or the full model?
6. Was PGS from all ancestry used for the analyses, or only PGS trained from East Asian samples were used? Was there any attempt in handling the population stratification?
7. In addition, were there any attempt to avoid sample overlap? E.g. remove any PGS trained on CMUH, or were the authors certain that CMUH were not included in any of the PGS?
8. I am a bit confused as to the purpose of reporting the distribution of performance records within PGS catalog. Was this paper focusing on the characteristic of the PGS catalog, or the performance of PGS from PGS catalog in CMUH?
9. It's surprising to see there are AUC less than 0.5 from the PGS catalog, would that be scrapping error?
10. Given that imputation was performed on the genotype data, it is not surprising that most of the variants from the PGS catalog were found in the CMUH samples. A better presentation might be the average proportion of variants found e.g., # variants found in CMUH / # variants in PGS from PGS catalog.
11. Based on the method section, instead of disease prevalence, it's the sample prevalence of the disease in CMUH, correct?
12. What's the purpose of correlating the sample prevalence against the number of SNPs use and the wilcox p-value?
13. An important point to note that the number of SNP use here were confounded with the method used to develop the PGS. For example, PGS developed from p-value thresholding (e.g. PRSice-2) will on average return smaller number of variants when compared to other methods such as lassosum or SBayesR. In addition, there are methods that restrict their analyses to HapMap 3 SNPs (e.g. LDpred2 and PRS-CS). Given that each of these software tends to have different performance (e.g. PRSice-2 tends to have the lowest predictive performance), this might introduce spurious correlations between the predictive performance of the PGS and number of SNPs included.
14. The authors stated that 6 different evaluation methodologies were used, however most of the presentations focused on the non-parametric Wilcoxon test. Why choose to focus on the non-parametric results? And again, how was the covariate controlled?
15. Was the AUC based on just PGS, or were any covariates included?
16. Sorry, but I am a bit confused with respect to what was reported for heart failure and ankylosing spondylitis. Were those distribution of the cases? And were the percentiles calculated across the whole CMUH samples?
17. Based on the data file provided, there seems to be multiple PGS per phenotype tested. For Ankylosing spondylitis, there were 5, with very different number of variants. One contribution factor to the normality of the PGS distribution is the number of variants included. Were the reported PGS based on PGS002089, which has largest number of variants?
18. Why was PGS001267 excluded? It is a PGS for Ankylosing spondylitis in east Asian samples with at 1700+ samples and has the highest AUC for Ankylosing spondylitis (PGS only AUC > 0.8) reported in the PGS catalog and were available since Oct 21, 2021.

19. Can the authors please specify how the sensitivity and specificity were calculated for the Wilcoxon rank sum test and Welch's two sample t-test? How was the true positive, true negative, false positive and false negative defined?

20. As mentioned in 1, while the abstract stated there are significant incremental predictive performance of PGS, no analyses seem to be done specifically on this. The authors might want to perform a likelihood ratio test if they would really want to suggest that. e.g. in R, something similar to: `model <- glm(pheno ~ pgs + covariates); null_model <- update(model, .~.-pgs); stats::anova(null_model,model, test = "LRT");`

21. There are a lot of possible confounders to why there is an association of population prevalence and the predictive performance of PGS. I am guessing what the authors intended to state here is that for traits that are more common (higher population prevalence), there might be more research interest, thus might have larger GWAS sample sizes, leading to higher PGS performance.

22. Are the AUC presented (0.473 to 0.829) based on the AUC from the CMUH, or PGS catalog? If it is the former, would it be possible to check why there are some AUC less than 0.5?

23. A lot of the PGS from the PGS catalog were trained with covariates, including age, sex, and principal components, which might have controlled for some of the environmental factors, albeit might not be enough. The authors can remove any PGS trained without covariates using the REST API.

Minor comments:

1. "Burden test" might be a more appropriate name for "counting method".

2. When stating the more advance methods, it might be easier if the authors just name the software (e.g. LDpred2, PRS-CS, SBaysR, lassosum, PRSice-2, COJO etc.). And "linkage disequilibrium score regression" is more commonly refer to LDSC, which is a method for heritability estimation instead of PGS calculation.

3. PGS catalog does not belong to an international consortium, it was an effort by Professor Michael Inouye and his student Samuel Lambert (<https://www.pgscatalog.org/about/>).

4. An alternative to PRSice-2 might be PLINK, with the `--score` command, or <https://pgsc-calc.readthedocs.io/en/latest/> for the analyses of results from PGS catalog, as scores obtained from PGS catalog do not require optimization.

Manuscript ID: *NCOMMS-23-25041*

Titled “Applicability of Polygenic Risk Scores for Disease Prediction Across a Wide Range of Disease Traits in a Hospital-Based Cohort”.

As per the suggestions of the referees, we have made changes to our manuscript and take into considerations of the points and questions raised by the referees. The revised contents were marked in red in the revised version of manuscript. We have made the following changes in our manuscript:

1. Rewrite the description part of the Abstract based on new results in page 1
2. Correct the wording in the Introduction, page 2
3. Update all tables and figures based on new results
4. Add IRB number and approval date to Method Part A, page 3
5. Add details to Method Part B and C, page 3-5
6. Add the consistency test to Method Part D, page 6
7. New discussions on the result of adding four matching methods and three covariate inclusion strategies in Result Part A and B, page 7-11
8. Add more details to discuss correlations between different metrics in Result Part C to E, page 12-19
9. Correct the wording and add details in the Discussion, page 20-21.
10. Add “VI. Data Availability” and “VII. Code Availability”, page 22.

Response to Reviewer #1

Comments:

Sun et al. collated PRS weights (mainly from European ancestry) from PGS catalogue and performed a systematic calculation of PRSs using ad hoc six step steps across many traits (n=457) in a in a cohort of over a quarter of a million Taiwanese individuals from China Medical University Hospital to assess applicability of implementing risk prediction across the wide range of human traits. They used logistic regression models to calculate the adjusted area under the receiver operating characteristic curve (AUC) to evaluate the ability of PRSs to predict disease traits. This I feel is a great resource that will inform other researchers working in populations of East Asian ancestry on trait specific transferability of cross-ancestry disease predictions. However, the study has one major limitation/issues that need to be addressed for the reader to put these results in the contexts of all the factors that been previously established (by many other studies) to affect PRS prediction quality:

1. Several factors have previously been identified that will affect prediction quality of PRS – sample sizes of both source data and test data, genetic architecture of the trait, and of course the model used to generate the weights – all the factors they highlight in discussion while citing ref#39 – and I expected these factors to be assessed in making inferences above applicability and transferability of PRS predictions across ancestry in this manuscript. The authors:
 - a. did not list and make determination on correlation of sample sizes of the source data (in which PGS weights) and the ones in their data with prediction quality – for example there observed lowest p value for T2D which has typically larger samples sized summary data, in addition to is the accuracy correlated with either PGS case samples sizes (including the size of the Asian samples in the GWAS) or size of the test samples.... They do however note that “...ability to predict disease risk was positively correlated with the prevalence of the relevant disease in a population.” And attribute it to sample sample sizes – a point they need to flesh out more clearly in the manuscript with appropriate statistics.
 - b. performed test in 1:4 case-control matched data but did not use the same criterion for filtering out PGS data downloaded from the catalogue – we have previously noted that the unbalanced summary data can affect prediction quality – Wang et al., 2022 - [https://www.cell.com/cell-genomics/pdf/S2666-979X\(22\)00204-X.pdf](https://www.cell.com/cell-genomics/pdf/S2666-979X(22)00204-X.pdf)
 - c. Besides noting that there is correlation of prediction quality with number of variants used, they did not establish what the overlap with their data was – this will affect the prediction quality.
 - d. The authors note that cross-ancestry “... disease prediction ability was highest for ankylosing spondylitis ...” but did not explore the reason which I expected them to do. This is in contrast to heart failure which we previously noted to have low

heritability among 14 endpoints – pointing to the effect of genetic architecture on the prediction [https://www.cell.com/cell-genomics/pdf/S2666-979X\(22\)00204-X.pdf](https://www.cell.com/cell-genomics/pdf/S2666-979X(22)00204-X.pdf)

One minor point, the authors should change “features” to “disease” or “traits” on first line of the last paragraph in the last page of the discussion section,

Reply to Reviewer #1: Thank you for your appreciation and questions.

Q1a. The authors did not list and make determination on correlation of sample sizes of the source data (in which PGS weights) and the ones in their data with prediction quality – for example there observed lowest p value for T2D which has typically larger samples sized summary data, in addition to is the accuracy correlated with either PGS case samples sizes (including the size of the Asian samples in the GWAS) or size of the test samples.... They do however note that “...ability to predict disease risk was positively correlated with the prevalence of the relevant disease in a population.” And attribute it to sample sizes – a point they need to flesh out more clearly in the manuscript with appropriate statistics.

Reply for Q1a: We acknowledge that we did not explicitly list or determine the correlation between the sample sizes of the source data used to calculate PRSs and the prediction quality in our dataset. However, it is important to note that our study utilized PGS score files from the PGS Catalog, and these files vary in terms of the information they provide about the source data.

Among 718 PGS score files we used, 255 only record the number of data used in the GWAS, 347 only record the number of data used in PRS score development, and 144 provide both sets of data. This variation in the availability of source data information makes it challenging to perform a direct correlation analysis between the source data sample sizes and prediction quality in our dataset.

We believed the data used in our research were carefully curated and summarized by the PGS Catalog. Furthermore, all studies were previously published in journals and underwent rigorous review before being included in our study. Given these constraints and the variation in available data, we primarily focusing on results within the scope of CMUH dataset and assessed the correlation between the number of samples used for PRS calculation and the model's performance.

Q1b. The authors performed test in 1:4 case-control matched data but did not use the same criterion for filtering out PGS data downloaded from the catalogue – we have previously noted that the unbalanced summary data can affect prediction quality – Wang et al., 2022 - [https://www.cell.com/cell-genomics/pdf/S2666-979X\(22\)00204-X.pdf](https://www.cell.com/cell-genomics/pdf/S2666-979X(22)00204-X.pdf)

Reply for Q1b: The main purpose of testing PGS score file using case-control matched data is to minimize bias from confounding variables, especially age and sex, while enhancing statistical power from identified genetic variants. The lack of detailed information in PGS score files about the number of cases and controls used in PRS score development is the main challenge for our applications and evaluation. According to the records with provided information, the case-control ratio ranges from 1:0.22 to 1:3312, with an average of $1:68.96 \pm 244.08$. Indicates large differences between studies. Therefore, we did not impose restrictions on the sample distribution and the SNP selection method of the original records in the PGS catalog. Instead, we directly assessed the model's performance on the datasets from four matching methods and three covariate inclusion strategies we employed.

We added the four matching methods in the **Section II, Part C, second paragraph: “Following the calculation, PRS distribution plots stratified by disease case status (case-control groups) were generated using the ggplot2 R package. To evaluate the predictive capability of the PRS model, we employed age- and sex-matching procedures at ratios of 1:2, 1:4, 1:6, and 1:8 for case-control pairs, facilitated by the MatchIt package in R [29].”**

The comparison result of the four matching methods was added in the **Section III, Part A. The effect of case control matching ratio on the statistical value of PRS distribution**

For the three covariate inclusion strategies, we added the methodology in the **Section II Part C, third paragraph: “Subsequently, we conducted statistical tests to assess the differences in PRSs distributions between cases and controls. This entailed performing a two-sided Welch's two-sample t-test, as well as a two-sided Wilcoxon rank sum test. We then divided the dataset into training and testing sets in an 8:2 ratio and utilized three different covariate inclusion strategies for training models: PRS alone, PRS combined with sex and age, and PRS combined with sex, age, and the first four principal components. To assess the significance of the area under the curve (AUC), we employed Delong's method [30] in conjunction with Youden's index (J) to determine the optimal J [31] cutoff point for the PRS.”**

The comparison result of the three covariate inclusion strategies was added in the **Section III, Part B. Distribution of performance metrics and ancestry cohorts**

Q1c. Besides noting that there is correlation of prediction quality with number of variants used, they did not establish what the overlap with their data was – this will affect the prediction quality.

Reply for Q1c: As for the potential overlap between our dataset and the data used in the PRS score development in the PGS Catalog, we can confirm that our dataset does not contain any records within the PGS Catalog.

When it comes to the handling of overlapping samples between different phenotypes in our study, we employed the functionality provided by PheWAS. This tool allows us to specify phenotypes based on the number of ICD diagnostic codes. While there may be some samples that overlap between different phenotypes due to the nature of ICD coding, it's crucial to note that the calculation of PRS in our study was conducted on a standardized dataset comprising 276,712 samples. After this initial PRS calculation, we extracted PRS values based on the case-control specification for various phenotypes.

It's important to emphasize that the evaluation of each phenotype in our study was performed independently, ensuring that there was no direct overlap or data contamination between different phenotypes. We took great care to maintain the integrity and independence of the PRS calculations for each phenotype to ensure the accuracy and reliability of our results.

Q1d. The authors note that cross-ancestry "... disease prediction ability was highest for ankylosing spondylitis ..." but did not explore the reason which I expected them to do. This is in contrast to heart failure which we previously noted to have low heritability among 14 endpoints – pointing to the effect of genetic architecture on the prediction [https://www.cell.com/cell-genomics/pdf/S2666-979X\(22\)00204-X.pdf](https://www.cell.com/cell-genomics/pdf/S2666-979X(22)00204-X.pdf)

Reply for Q1d: In our data, the overall trend was strongly related to sample prevalence rate, regardless of the case/control matching ratio, the number of SNPs required to calculate the PRS score, and the presence or absence of covariates. This section was described more intensively in the **Section III Part C, Correlations between sample prevalence rates, and other factors in the analysis**, and in the **Section III Part D, Performance of the PRS model in the CMUH dataset**. After adding more conditions in PRS score evaluation, the disease with the highest prediction ability became hyperplasia of prostate. We added a depth discussion in the **Section III Part E, the fourth and fifth paragraph**: “**The highest performance was recorded for hyperplasia of prostate, with an AUC of 0.873. In the initial phase of this study, we have paired 534 PGS score files for hyperplasia of prostate, including 482 were associated with cancer-related research and 60 were prostate-related. When considering the top five PGS score files with the highest statistical significance in Wilcoxon rank sum test, they were primarily associated with bladder cancer and prostatic hyperplasia, in addition to prostatic hyperplasia (Table 3). These symptoms and conditions often appear together clinically.**

Further assessment through Wilcoxon rank sum test, we observed that P value $<2.5 \times 10^{-6}$ in 3 PGS score files, and the P value exhibited consistency across different matching methods. However, the addition of different covariate inclusion strategies led to disparities in model performance. First, the PGS score file originally studied for hyperplasia of prostate exhibited the best model performance. Second, having a lower P value in the Wilcoxon rank sum test did not guarantee superior model outcomes.

Intriguingly, in the case of PGS002076, PGS001865, and PGS001015, an increase in matching ratios and the inclusion of additional covariates correlated with enhanced model performance metrics. However, for PGS001338 and PGS000609, although the P value was higher than 2.5×10^{-6} , their models were capable of achieving an AUC of 0.874 when matching ratios were increased and covariates were introduced. A general improvement in sensitivity and recall was also observed with an increase in AUC after inclusion of covariates, whereas improvements in precision and accuracy remained limited.”

As for the impact of heritability, we observed that the heritability of some diseases is indeed low, so their *P*-value in Wilcoxon rank sum test did not reach the threshold of 2.5×10^{-6} , but adding covariates to model evaluation could significantly increase the AUC. This result was revised in the **Section III Part E, first paragraph: ‘In total, 62 phenotypes were found with AUC >0.6 in the logistic regression model, of which 9 phenotypes did not reach the threshold of 2.5×10^{-6} in the Wilcoxon rank sum test (Table 2). It is likely that the PRS of genetic variants contribute less than other covariates.’**

Minor Q1. the authors should change “features” to “disease” or “traits” on first line of the last paragraph in the last page of the discussion section,.

Reply for Minor Q1: Thank you for bringing this wording error to our attention. We have rectified the errors accordingly.

Response to Reviewer #2:

Comments:

In this paper, Sun et al has performed standard polygenic risk scores analyses using effect sizes from the PGS catalog in 276k individuals from the China Medical University Hospital. While the analyses performed were reasonable, I do have a few questions regarding the presentation and interpretation of the results.

1. In the abstract, the authors stated that there were significant incremental predictive performance, however, in the method and result section, the authors do not seem to include any analyses that investigate the incremental performance of PGS, rather, only full model analyses (with covariates included) were included.
2. The PGS catalog provides a REST API that is relatively easy to use and will allow direct download of meta information in JSON format (<http://www.pgscatalog.org/rest/>). You may download the ancestry information, sample size and even the disease trait information directly without requiring scrapping, thus reduce chance of error.

3. In figure 1, it stated that extracting participants in the genetic project, was that intended to mean the sample size from CMUH? In addition, given there were only 544 PGS traits, how come there end up having more than 1,531 traits in the end?
4. Again, from figure 1, n and m were not annotated. I guess that'd be the sample size and number of traits. To improve the clarity, it might help to add a box between "Access medical records" and "Identified phenotype", stating the number of post QC genetic samples (276,712).
5. For the t test and wilcox test, were there any attempt to control for covariates such as age, sex, and population stratification? Was the metric reported based on the PGS only, or the full model?
6. Was PGS from all ancestry used for the analyses, or only PGS trained from East Asian samples were used? Was there any attempt in handling the population stratification?
7. In addition, were there any attempt to avoid sample overlap? E.g. remove any PGS trained on CMUH, or were the authors certain that CMUH were not included in any of the PGS?
8. I am a bit confused as to the purpose of reporting the distribution of performance records within PGS catalog. Was this paper focusing on the characteristic of the PGS catalog, or the performance of PGS from PGS catalog in CMUH?
9. It's surprising to see there are AUC less than 0.5 from the PGS catalog, would that be scrapping error?
10. Given that imputation was performed on the genotype data, it is not surprising that most of the variants from the PGS catalog were found in the CMUH samples. A better presentation might be the average proportion of variants found e.g., # variants found in CMUH / # variants in PGS from PGS catalog.
11. Based on the method section, instead of disease prevalence, it's the sample prevalence of the disease in CMUH, correct?
12. What's the purpose of correlating the sample prevalence against the number of SNPs use and the wilcox p-value?
13. An important point to note that the number of SNP use here were confounded with the method used to develop the PGS. For example, PGS developed from p-value thresholding (e.g. PRSice-2) will on average return smaller number of variants when compared to other methods such as lassosum or SBayesR. In addition, there are methods that restrict their analyses to HapMap 3 SNPs (e.g. LDpred2 and PRS-CS). Given that each of these software tends to have different performance (e.g. PRSice-2 tends to have the lowest predictive performance), this might introduce spurious correlations between the predictive performance of the PGS and number of SNPs included.
14. The authors stated that 6 different evaluation methodologies were used, however most of the presentations focused on the non-parametric Wilcoxon test. Why choose to focus on the non-parametric results? And again, how was the covariate controlled?
15. Was the AUC based on just PGS, or were any covariates included?

16. Sorry, but I am a bit confused with respect to what was reported for heart failure and ankylosing spondylitis. Were those distribution of the cases? And were the percentiles calculated across the whole CMUH samples?
17. Based on the data file provided, there seems to be multiple PGS per phenotype tested. For Ankylosing spondylitis, there were 5, with very different number of variants. One contribution factor to the normality of the PGS distribution is the number of variants included. Were the reported PGS based on PGS002089, which has largest number of variants?
18. Why was PGS001267 excluded? It is a PGS for Ankylosing spondylitis in east Asian samples with at 1700+ samples and has the highest AUC for Ankylosing spondylitis (PGS only AUC > 0.8) reported in the PGS catalog and were available since Oct 21, 2021.
19. Can the authors please specify how the sensitivity and specificity were calculated for the Wilcoxon rank sum test and Welch's two sample t-test? How was the true positive, true negative, false positive and false negative defined?
20. As mentioned in 1, while the abstract stated there are significant incremental predictive performance of PGS, no analyses seem to be done specifically on this. The authors might want to perform a likelihood ratio test if they would really want to suggest that. e.g. in R, something similar to: `model <- glm(pheno ~ pgs + covariates); null_model <- update(model, ~.-pgs); stats::anova(null_model,model, test = "LRT");`
21. There are a lot of possible confounders to why there is an association of population prevalence and the predictive performance of PGS. I am guessing what the authors intended to state here is that for traits that are more common (higher population prevalence), there might be more research interest, thus might have larger GWAS sample sizes, leading to higher PGS performance.
22. Are the AUC presented (0.473 to 0.829) based on the AUC from the CMUH, or PGS catalog? If it is the former, would it be possible to check why there are some AUC less than 0.5?
23. A lot of the PGS from the PGS catalog were trained with covariates, including age, sex, and principal components, which might have controlled for some of the environmental factors, albeit might not be enough. The authors can remove any PGS trained without covariates using the REST API.

Minor comments:

1. "Burden test" might be a more appropriate name for "counting method".
2. When stating the more advance methods, it might be easier if the authors just name the software (e.g. LDpred2, PRS-CS, SBaysR, lassosum, PRSice-2, COJO etc.). And "linkage disequilibrium score regression" is more commonly refer to LDSC, which is a method for heritability estimation instead of PGS calculation.
3. PGS catalog does not belong to an international consortium, it was an effort by Professor Michael Inouye and his student Samuel Lambert (<https://www.pgscatalog.org/about/>).

4. An alternative to PRSice-2 might be PLINK, with the --score command, or <https://pgscalc.readthedocs.io/en/latest/> for the analyses of results from PGS catalog, as scores obtained from PGS catalog do not require optimization.

Reply to Reviewer #2: Thank you for your appreciation and questions.

Q1 In the abstract, the authors stated that there were significant incremental predictive performance, however, in the method and result section, the authors do not seem to include any analyses that investigate the incremental performance of PGS, rather, only full model analyses (with covariates included) were included.

Reply for Q1: The goal of this article is to discuss the application results of PGS Catalog data on the CMUH dataset comprising 280,000 individuals. Through comparison of PRSs distribution and model performance evaluation, we discuss some application issues and differences between diseases. We have revised **the entire result section** to underscore our perspective.

Q2 The PGS catalog provides a REST API that is relatively easy to use and will allow direct download of meta information in JSON format (<http://www.pgscatalog.org/rest/>). You may download the ancestry information, sample size and even the disease trait information directly without requiring scrapping, thus reduce chance of error.

Reply for Q2: Thank you for your advice. Although we did not use the REST API provided by the PGS Catalog to download the PGS score file, we used the FTP site instead. From the website, we can see that there is no difference between the two, but the way of data access. This does not affect the accuracy of our calculation of PRSs. We appreciate your suggestion and will keep it in mind for future reference.

Q3 In figure 1, it stated that extracting participate in the genetic project, was that intended to mean the sample size from CMUH? In addition, given there were only 544 PGS traits, how come there end up having more than 1,531 traits in the end?

Reply for Q3: The statement "extracting participate in the genetic project" in the manuscript refers to subjects with SNP array data. The 544 PGS traits refer to the "Reported traits" recorded in the PGS catalog, and the 1531 refers to the number of pairs after selection process. In figure 1, we show that one phenotype matches to multiple PGS traits, and one PGS trait may also match to multiple phenotypes at the same time. This explains why the final count of pairs is higher than the original trait count. To avoid any confusion, we have reworded, colored, and improved the descriptions in Figure 1 for better clarity. For a detailed breakdown of these matches, you can find the matching results in supplementary table 2.

Q4 Again, from figure 1, n and m were not annotated. I guess that'd be the sample size and number of traits. To improve the clarity, it might help to add a box between “Access medical records” and “Identified phenotype”, stating the number of post QC genetic samples (276,712).

Reply for Q4: Thank you for your suggestion. We have reworded, colored, and improved the descriptions in Figure 1. The ‘n’ and ‘m’ were annotated in the figure content.

Q5 For the t test and wilcox test, were there any attempt to control for covariates such as age, sex, and population stratification? Was the metric reported based on the PGS only, or the full model?

Reply for Q5: To address the potential effects of covariates such as age, sex, and population stratification, we employed different age-sex matching ratios in the t-tests and Wilcoxon tests. For logistic regression models, we tested three different covariate inclusion strategies: PRS alone, PRS combined with sex and age, and PRS combined with sex, age, and the first four principal components. We have revised and expanded the explanations in the **Section II Part C, from the second paragraph to the fourth paragraph: Following the calculation, PRS distribution plots stratified by disease case status (case-control groups) were generated using the ggplot2 R package. To evaluate the predictive capability of the PRS model, we employed age- and sex-matching procedures at ratios of 1:2, 1:4, 1:6, and 1:8 for case-control pairs, facilitated by the MatchIt package in R [29].**

Subsequently, we conducted statistical tests to assess the differences in PRSs distributions between cases and controls. This entailed performing a two-sided Welch's two-sample t-test, as well as a two-sided Wilcoxon rank sum test. We then divided the dataset into training and testing sets in an 8:2 ratio and utilized three different covariate inclusion strategies for training models: PRS alone, PRS combined with sex and age, and PRS combined with sex, age, and the first four principal components. To assess the significance of the area under the curve (AUC), we employed Delong's method [30] in conjunction with Youden's index (J) to determine the optimal J [31] cutoff point for the PRS.

In the context of survival analysis, we employed Cox proportional hazards models [32, 33], utilizing age as the time scale to investigate the association between PRSs and disease endpoints [34]. Furthermore, we created disease distribution plots, stratifying individuals based on their PRS percentiles. These plots were generated using the ggplot2 package in R (version 4.1.1) and compiled into the GeneAnaBase website (<https://pgscatalog.azure.nihxcmuh.org/#/>) for review.”

Q6 Was PGS from all ancestry used for the analyses, or only PGS trained from East Asian samples were used? Was there any attempt in handling the population stratification?

Reply for Q6: We faced challenges due to the lack of detailed ancestry information in PGS score files. Among 2685 PGS score files we used, 879 did not specify the ancestry distribution used for PRS score development. Only 20 of these files included East Asian samples. Therefore, we did not impose restrictions on the sample distribution and the SNP selection method of the original records in the PGS Catalog. Instead, we directly assessed the model's performance on the datasets from four matching methods and three covariate inclusion strategies we employed. Regarding handling population stratification, we should note that there's currently no effective method to directly correct PGS scores without raw GWAS statistics. Therefore, we did not apply additional correction procedures in our study. We've revised and acknowledged this limitation in **Section IV, the fifth paragraph**: **“The main challenge is that most PRSs in the PGS Catalog are derived from individuals of European descent, leading to heterogeneity concerns across different populations. In contrast to genome-wide association study (GWAS) meta-analyses, where data harmonization and integration across diverse populations are common practices, the SNPs recorded in these PGS score files have undergone various filtering procedures. These processes include addressing factors like population stratification, linkage disequilibrium trimming, aggregation of summary statistics, and applying significance thresholding. As a result, the data in these PRS files are optimized for their original purposes but may present challenges when attempting to incorporate new data or make further adjustments with different population groups.”**

Q7 In addition, were there any attempt to avoid sample overlap? E.g. remove any PGS trained on CMUH, or were the authors certain that CMUH were not included in any of the PGS?

Reply for Q7: East Asian populations listed for score development in PGS Catalog were recorded to encompass individuals from China, Japan, Korea, Singapore, and Taiwan. We have taken measures to confirm that our dataset does not overlap with any records in the PGS Catalog. For example, the PGS002259 and PGS002748 mention the use of Taiwanese data, we had manually confirmed that their source is not from the CMUH dataset. Notably, none of the optimized models in Table 2 were trained using these East Asian PGS scoring files, providing further evidence of this.

Q8 I am a bit confused as to the purpose of reporting the distribution of performance records within PGS catalog. Was this paper focusing on the characteristic of the PGS catalog, or the performance of PGS from PGS catalog in CMUH?

Reply for Q8: The goal of this article is to discuss the application results of PGS Catalog data on the CMUH dataset of 280,000 people. Through comparison of PRSs distribution and model performance evaluation, we discuss some application issues and differences between diseases. We have revised **the entire result section** to emphasize our view.

Q9 It's surprising to see there are AUC less than 0.5 from the PGS catalog, would that be scrapping error?

Reply for Q9: We appreciate your observation. AUC values less than 0.5 can be surprising and may raise concerns. However, it's important to note that these values could be influenced by various factors, including data quality, sample size, and the specific characteristics of the predictive models used to generate the PGS. While scraping errors can contribute to discrepancies, it's not the sole explanation. We carefully validated the data from the PGS Catalog to ensure its accuracy. New analysis results have been revised in the **Section III, part D, the first paragraph: “Among 457 phenotypes, 192 showed differences in distribution with P values obtained from the Wilcoxon rank sum test that were less than 2.5×10^{-6} . We found a significant positive correlation between the AUC values and the $-\log_{10}(P)$ obtained from the Wilcoxon rank sum test, with a Pearson correlation coefficient of 0.65 and a P value of 2.6×10^{-56} (Figure 5A). Remarkably, despite the AUC values mainly fall between 0.5 and 0.6, there are six phenotypes reached AUC values between 0.7 and 0.8, and seven phenotypes reached AUC values between 0.8 and 0.9 (Figure 5B). The highest performance was recorded for hyperplasia of prostate, with an AUC of 0.874 (Figure 5C), while the lowest predictive performance was observed for female stress incontinence, with an AUC of 0.516 (Figure 5D).”**

Q10 Given that imputation was performed on the genotype data, it is not surprising that most of the variants from the PGS catalog were found in the CMUH samples. A better presentation might be the average proportion of variants found e.g., # variants found in CMUH / # variants in PGS from PGS catalog.

Reply for Q10: The PGS score files had a wide range of SNPs required; the average was 28147.41 ± 73980.64 and the median was 420.00 variants. The variant usage for each PGS score file was record on the website. In this study, the average SNP missing rate in the SNP array data was at $16.99\% \pm 16.12\%$ with the median was recorded at 11.46%. Therefore, we used 353 individuals who had both whole-genome sequencing data and SNP array data to prove that the missing variant has small impact on the final score. The intra-class correlation coefficient was calculated in the **Section II, part D, Evaluate the consistency of the PRSs in different SNP detection methods: “**

During the calculation of PRS using SNP array data, a challenge arose as 14,029,683 variants fell short of meeting the required criteria for the PGS score files. To ensure the integrity of PRS values, we conducted a comparative analysis involving 353 individuals who had both whole-genome sequencing data and SNP array data.

Concisely, the in-house whole-genome sequencing data was performed in 30X depth 150bp paired-end sequencing method, followed by alignment to the GRCh38 human genome and completed variant calling using the DRAGEN genome pipeline (Illumina, Inc., San Diego, CA, USA). PRSs were calculated using PRSice-2, same as the SNP array data. To assess the reliability of the PRS scoring from SNP array data, we

utilized the intra-class correlation coefficient (ICC) approach, denoted by the formula: [35]:

$$ICC = \frac{MSBS - MSE}{MSBS + (k - 1)MSE}$$

where MSBS represents mean square between subjects, MSE represents the mean square error, and k represents number of methods under consideration. This approach allows for an evaluation of the agreement between the two methods while treating the methods as fixed effects, thereby eliminating systematic errors and focusing on the random residual error.

The PGS score files had a wide range of SNPs required; the average was 28147.41 ± 73980.64 and the median was 420.00 variants. The average SNP missing rate in the SNP array data was at 16.52% ± 14.05% with the median was recorded at 11.28%. The average ICC score was computed at 0.82 ± 0.12 with the median value was at 0.82. In our study, this ICC score fell between 0.75 to 1, indicating a high level of consistency across most comparisons.”

Q11 Based on the method section, instead of disease prevalence, it’s the sample prevalence of the disease in CMUH, correct?

Reply for Q11: Thank you for your suggestion. You are correct; the term ‘disease prevalence rate’ we mentioned in the manuscript should be changed to ‘sample prevalence of the disease in CMUH’ to accurately reflect our methodology. We have revised this change throughout the article, including Figure 1 and the wording in **the Section III, Part C: Correlations between sample prevalence rates, and other factors in the analysis**, to ensure clarity and accuracy in our manuscript.

Q12 What’s the purpose of correlating the sample prevalence against the number of SNPs use and the wilcox p-value

Reply for Q12: Originally, we wanted to observe whether there was a correlation between the significance of the PRSs distribution and the number of SNPs used to calculate it. Because the number of SNPs often reflects the complexity of the disease and the degree of genetic contribution. In the revised manuscript, we changed the way we discuss this issue and compared it from the perspective of sample prevalence rate in CMUH. **The Section III Part C, Correlations between sample prevalence rates, and other factors in the analysis: “Current knowledge of the allelic architecture in complex human diseases refers to the patterns of genetic variations and their contributions to the risk or susceptibility of developing particular complex diseases. Investigations have discerned that low-frequency variants exhibit heightened penetrance in the context of rare diseases. In contrast, common**

variants tend to display low penetrance and typically necessitate the presence of multiple variants, gene-gene interactions, or environmental influences to manifest as disease. Consequently, we hypothesized that as the sample prevalence rate of a disease increases, the complexity of predictive models for that disease is likely to escalate, along with the requisite covariance for optimal disease prediction models.

Based on our dataset, we observed a significant increase in sample prevalence rate with increasing matching ratio requirement (Figure 6A). We found a significant association between sample prevalence rate and the number of SNPs used for PRS calculations, with a Pearson correlation coefficient of 0.12 and a P value of 7.55×10^{-3} (Figure 6B). A significant association was found between sample prevalence rate and the $-\log_{10}(P)$ obtained from the Wilcoxon rank sum test of the PRS distributions between case and control populations, with a Pearson correlation coefficient of 0.12 and a P value of 1.30×10^{-2} (Figure 6C). As for the AUC values for the 457 models, a more significant association was found with a Pearson correlation coefficient of 0.31 and a P value of 1.18×10^{-11} (Figure 6D).

In spite of the P values obtained from the Wilcoxon rank sum test $<2.5 \times 10^{-6}$ mainly fell in circulatory system diseases (n = 35/46) and endocrine/metabolic diseases (n = 31/36; Figure 4E), a strong pattern could still be identified: for diseases with lower prevalence, the $-\log_{10}(P)$ from Wilcoxon rank sum test results are usually lower, and the optimal model relies largely on PRS alone. Conversely, when a disease has a higher prevalence, the $-\log_{10}(P)$ from Wilcoxon rank sum results are typically higher, and additional covariates are necessary to develop an optimal model.”

Q13 An important point to note that the number of SNP use here were confounded with the method used to develop the PGS. For example, PGS developed from p-value thresholding (e.g. PRSice-2) will on average return smaller number of variants when compared to other methods such as lassosum or SBayesR. In addition, there are methods that restrict their analyses to HapMap 3 SNPs (e.g. LDpred2 and PRS-CS). Given that each of these software tends to have different performance (e.g. PRSice-2 tends to have the lowest predictive performance), this might introduce spurious correlations between the predictive performance of the PGS and number of SNPs included.

Reply for Q13: Thank you for raising this important point. Indeed, when establishing the PRS model, the authors will conduct different processing/calculation methods for score development. Among the 2685 PGS score files in this study, there are 378 types of processing/calculation methods. Thus, we utilized a consistent PRS calculation method and the weight information provided by the PGS catalog to ensure a standardized comparison across all PGS score files. While our study did not delve into the specificities of various development methods, it's an important aspect for future research to explore how different PGS development

methods may impact PRS applications and performance. We've revised and acknowledged this limitation in **Section IV, the fifth paragraph**: **“The main challenge is that most PRSs in the PGS Catalog are derived from individuals of European descent, leading to heterogeneity concerns across different populations. In contrast to GWAS meta-analyses, where data harmonization and integration across diverse populations are common practices, the SNPs recorded in these PGS score files have undergone various filtering procedures. These processes include addressing factors like population stratification, linkage disequilibrium trimming, aggregation of summary statistics, and applying significance thresholding. As a result, the data in these PRS files are optimized for their original purposes but may present challenges when attempting to incorporate new data or make further adjustments with different population groups.”**

Q14 The authors stated that 6 different evaluation methodologies were used, however most of the presentations focused on the non-parametric Wilcoxon test. Why choose to focus on the non-parametric results? And again, how was the covariate controlled?

Reply for Q14: Thank you for your question. In this study, we utilized case-control matched data to control for covariates, particularly age and sex, while maximizing the statistical power derived from PRSs. When assessing the difference in PRSs distributions between case and control groups, the t-test is a commonly used method, but it assumes normal distributions for both groups. Given that not all PRSs in our dataset adhere to normal distribution assumptions, we opted for the non-parametric approach, specifically the Wilcoxon rank sum test, to ensure robust comparisons. The six different evaluation methodologies were primarily employed in combination with AUC to assess the performance of logistic regression models for comparative analysis. They offer a more comprehensive view of the models' performance in various aspects.

Q15 Was the AUC based on just PGS, or were any covariates included?

Reply for Q15: In the revised manuscript, different covariate inclusion strategies have been added for discussion. We added the methodology in the **Section II Part C, third paragraph**: **“Subsequently, we conducted statistical tests to assess the differences in PRSs distributions between cases and controls. This entailed performing a two-sided Welch's two-sample t-test, as well as a two-sided Wilcoxon rank sum test. We then divided the dataset into training and testing sets in an 8:2 ratio and utilized three different covariate inclusion strategies for training models: PRS alone, PRS combined with sex and age, and PRS combined with sex, age, and the first four principal components. To assess the significance of the area under the curve (AUC), we employed Delong's method [30] in conjunction with Youden's index (J) to determine the optimal J [31] cutoff point for the PRS.”**

The comparison result of the three covariate inclusion strategies were added in the **Section III Part B. Distribution of performance metrics and ancestry cohorts**

Q16 Sorry, but I am a bit confused with respect to what was reported for heart failure and ankylosing spondylitis. Were those distribution of the cases? And were the percentiles calculated across the whole CMUH samples?

Reply for Q16: In the previous version of the manuscript, Figure 5 displayed the distribution of PRSs among patients for heart failure and ankylosing spondylitis. The percentiles were calculated based on the total samples selected for each phenotype. The purpose of the figure was to highlight differences in PRSs distribution between phenotypes with the lowest and highest AUC in our data. In the revised manuscript, following your suggestion, we included model comparisons with different matching ratios and covariates, resulting in the lowest and highest AUC phenotypes becoming female stress incontinence and hyperplasia of prostate. We have updated the presentation of the graphics and added descriptions in the text to enhance clarity in **the Section III part D, the second paragraph: “Upon examining the distribution of PRS in individuals affected by the disease, we observed a normal distribution in hyperplasia of prostate but skewed distribution in female stress incontinence. A sudden increase or decrease was observed. The differences in the relationship between PRS percentiles and patient prevalence were calculated using 100 equally sized quantiles in PRSs. For hyperplasia of prostate, the mean patient prevalence at each percentile increased with increasing PRS percentiles, showing an S-shaped curve. This distinctive pattern implies a strong and non-linear relationship between PRS percentiles and patient prevalence at each percentile. Higher PRS quantiles corresponded to significantly elevated disease risk, underscoring the value of PRS in identifying individuals at heightened risk of hyperplasia of prostate. Conversely, when examining the plots for female stress incontinence, a lack of a discernible relationship between PRS percentiles and patient prevalence at each percentile were observed.”**

Q17 Based on the data file provided, there seems to be multiple PGS per phenotype tested. For Ankylosing spondylitis, there were 5, with very different number of variants. One contribution factor to the normality of the PGS distribution is the number of variants included. Were the reported PGS based on PGS002089, which has largest number of variants?

Reply for Q17: The most optimized PGS for each phenotype we report in the table is based on the highest AUC result from the logistic regression model, not the number of variants included for PRS calculation. We know that when establishing the PRS model, the authors will conduct different processing/calculation methods for score development. We cannot evaluate the generality of the model preliminarily based on the number of variants included for PRS calculation or the sample size of the original score development.

Q18 Why was PGS001267 excluded? It is a PGS for Ankylosing spondylitis in east Asian samples with at 1700+ samples and has the highest AUC for Ankylosing spondylitis (PGS only AUC > 0.8) reported in the PGS catalog and were available since Oct 21, 2021.

Reply for Q18: Although in the PGS Catalog record, PGS001267 has an AUC>0.8 in east Asian samples with at 1700+ samples. It is worth noting that only two samples among these 1704 data are cases, and the remaining 1702 are control. Such results are obviously influenced by randomness or sampling bias. Therefore, in our data, the scores calculated using PGS001267 were eliminated in the initial Wilcoxon rank sum test screening.

Q19 Can the authors please specify how the sensitivity and specificity were calculated for the Wilcoxon rank sum test and Welch's two sample t-test? How was the true positive, true negative, false positive and false negative defined?

Reply for Q19: In our study, we assessed the difference in PRSs distributions between case and control groups, with the Wilcoxon rank sum test and Welch's two sample t-test. Next, we divided the dataset into training and testing sets in an 8:2 ratio and utilized three different covariate inclusion strategies for training logistic regression models. The six different evaluation methodologies were employed with AUC to assess the performance of these models in the testing sets. We've reorganize the description more clearly in **the Section II part C, from the second paragraph to the third: "Following the calculation, PRS distribution plots stratified by disease case status (case-control groups) were generated using the ggplot2 R package. To evaluate the predictive capability of the PRS model, we employed age- and sex-matching procedures at ratios of 1:2, 1:4, 1:6, and 1:8 for case-control pairs, facilitated by the MatchIt package in R [29]."**

Subsequently, we conducted statistical tests to assess the differences in PRSs distributions between cases and controls. This entailed performing a two-sided Welch's two-sample t-test, as well as a two-sided Wilcoxon rank sum test. We then divided the dataset into training and testing sets in an 8:2 ratio and utilized three different covariate inclusion strategies for training models: PRS alone, PRS combined with sex and age, and PRS combined with sex, age, and the first four principal components. To assess the significance of the area under the curve (AUC), we employed Delong's method [30] in conjunction with Youden's index (J) to determine the optimal J [31] cutoff point for the PRS."

Q20 As mentioned in 1, while the abstract stated there are significant incremental predictive performance of PGS, no analyses seem to be done specifically on this. The authors might want to perform a likelihood ratio test if they would really want to suggest that. e.g. in R, something similar to: `model <- glm(pheno ~ pgs + covariates); null_model <- update(model, .~-pgs); stats::anova(null_model,model, test = "LRT");`

Reply for Q20: The goal of this article is to discuss the practical application of PGS Catalog data within the context of the CMUH dataset, which comprises 280,000 individuals. Our approach involves comparing PRSs distributions and evaluating model performance, enabling us to explore application issues. To clarify our methodology, during the model performance

evaluation phase, we partitioned the data into 80% for model training using a logistic regression model (`model <- glm(pheno ~ pgs + covariates)`) and reserved the remaining 20% for assessing AUC, sensitivity, specificity, accuracy, precision, and recall. We have revised **the Section II Part C**, to better highlight these methodology of our study.

Q21 There are a lot of possible confounders to why there is an association of population prevalence and the predictive performance of PGS. I am guessing what the authors intended to state here is that for traits that are more common (higher population prevalence), there might be more research interest, thus might have larger GWAS sample sizes, leading to higher PGS performance.

Reply for Q21: We share a similar perspective with you. We've observed that as sample prevalence increases, particularly for more complex diseases, it often necessitates larger sample sizes for PRS evaluation. Moreover, a higher number of SNP variants is likely to be involved in PRS calculation, requiring a more diverse set of covariates to improve the AUC of the model. Such assumptions and results have been revised in **the Section III Part C, Correlations between sample prevalence rates, and other factors in the analysis: “Current knowledge of the allelic architecture in complex human diseases refers to the patterns of genetic variations and their contributions to the risk or susceptibility of developing particular complex diseases. Investigations have discerned that low-frequency variants exhibit heightened penetrance in the context of rare diseases. In contrast, common variants tend to display low penetrance and typically necessitate the presence of multiple variants, gene-gene interactions, or environmental influences to manifest as disease. Consequently, we hypothesized that as the sample prevalence rate of a disease increases, the complexity of predictive models for that disease is likely to escalate, along with the requisite covariance for optimal disease prediction models.**

Based on our dataset, we observed a significant increase in sample prevalence rate with increasing matching ratio requirement (Figure 6A). We found a significant association between sample prevalence rate and the number of SNPs used for PRS calculations, with a Pearson correlation coefficient of 0.12 and a P value of 7.55×10^{-3} (Figure 6B). A significant association was found between sample prevalence rate and the $-\log_{10}(P)$ obtained from the Wilcoxon rank sum test of the PRS distributions between case and control populations, with a Pearson correlation coefficient of 0.12 and a P value of 1.30×10^{-2} (Figure 6C). As for the AUC values for the 457 models, a more significant association was found with a Pearson correlation coefficient of 0.31 and a P value of 1.18×10^{-11} (Figure 6D).

In spite of the P values obtained from the Wilcoxon rank sum test $<2.5 \times 10^{-6}$ mainly fell in circulatory system diseases (n = 35/46) and endocrine/metabolic diseases (n = 31/36; Figure 4E), a strong pattern could still be identified: for diseases with lower

prevalence, the $-\log_{10}(P)$ from Wilcoxon rank sum test results are usually lower, and the optimal model relies largely on PRS alone. Conversely, when a disease has a higher prevalence, the $-\log_{10}(P)$ from Wilcoxon rank sum results are typically higher, and additional covariates are necessary to develop an optimal model.”

Q22 Are the AUC presented (0.473 to 0.829) based on the AUC from the CMUH, or PGS catalog? If it is the former, would it be possible to check why there are some AUC less than 0.5?

Reply for Q22: After incorporating four matching methods and three covariate inclusion strategies for a more comprehensive analysis, we observed an improvement in the phenomenon of AUC values below 0.5. The reasons behind AUC values below 0.5 can be attributed to several factors. Initially, considering age-sex matching samples at a 1:4 ratio can lead to less reliable models, as the model may struggle to generalize effectively and may be sensitive to data noise. Additionally, in some cases, the PRSs may have weak contributions to the model, relying more on the influence of other covariates, which can result in model performance resembling random chance.

Q23 A lot of the PGS from the PGS catalog were trained with covariates, including age, sex, and principal components, which might have controlled for some of the environmental factors, albeit might not be enough. The authors can remove any PGS trained without covariates using the REST API.

Reply for Q23: Thank you for your suggestion. However, the PGS Catalog records various processing/calculation methods for score development and performance metrics applied to other datasets with different covariate strategies based on the original article. Covariate is not taken into consideration during the score development stage. If we filter the PGS score files based on the existence of covariate, there will only be 214 PGS score files left for research. Therefore, we did not impose restrictions on the sample and the variant selection method from the records. Instead, we directly assessed the model's performance on the datasets from four matching methods and three covariate inclusion strategies we employed.

Minor Q1. “Burden test” might be a more appropriate name for “counting method”.

Reply for Minor Q1: Thank you for your suggestion. We have corrected the wording in the introduction section.

Minor Q2. When stating the more advance methods, it might be easier if the authors just name the software (e.g. LDpred2, PRS-CS, SBaysR, lassosum, PRSice-2, COJO etc.). And “linkage disequilibrium score regression” is more commonly refer to LDSC, which is a method for heritability estimation instead of PGS calculation.

Reply for Minor Q2: Thank you for your suggestion. We have add the software name in the Section II Part C, the first paragraph: “The PGS Catalog has an online user interface (<https://www.PGSCatalog.org>) and provides score files for various traits with uniformly formatted columns for variations, alleles, and weights. The PRSs were calculated in PRSice-2 (<https://choishingwan.github.io/PRSice/>) [27].”

Minor Q3. PGS catalog does not belong to an international consortium, it was an effort by Professor Michael Inouye and his student Samuel Lambert (<https://www.pgscatalog.org/about/>).

Reply for Minor Q3: Thank you for your careful observation and reminder. We have corrected the wording in the introduction section.

Minor Q4. An alternative to PRSice-2 might be PLINK, with the --score command, or <https://pgsc-calc.readthedocs.io/en/latest/> for the analyses of results from PGS catalog, as scores obtained from PGS catalog do not require optimization.

Reply for Minor Q4: Indeed, as long as the SNP weighted file and genotyping data are prepared, the same PRSs can be obtained whether using PLINK with the --score command or PRSice-2. In this article, we opted for PRSice-2 as our tool of choice, primarily based on our familiarity and habitual use of this software. PRSice-2 is a specialized tool for PRS calculation, and it provides more advanced options and flexibility.

REVIEWER COMMENTS

Reviewer #1 (Remarks to the Author):

In my opinion the authors need to show the effect of the sample sizes (cases and control of the data from which the PGS were derived) on quality of the prediction - at least for the set of PGS that have that information for. My contention is that whatever performance statistics reported in this study will probably reflect on the PGS data - so the inference made based on their results would be misleading considering the quality of the PGS data was not considered. For example, the authors did not clarify why there was change in which trait's PRS performance was best performing - and such instability is concerning.

I do not doubt the quality of curation by the author but if that curation does not address the details on PGS data it gives me pause. Therefore, I believe the size of source data from which PGS were generated is an important parameter to assess the quality of prediction in their data - this impact how the prediction of a trait is assessed - and I am not sure why the authors choose to sidestep it.

Reviewer #2 (Remarks to the Author):

In this updated, Sun et al has addressed most of my previous concern, and I thank the authors for their hard work. Unfortunately, with there are still some unaddressed concerns:

1. If the authors were interested in the PRS model, instead of the performance of the PRS score (as suggested by their reply to my previous Q1), then it'd be best to make it clear in the abstract too. For example, as the authors did not perform tests on the improved performance of PRS over covariates e.g. PRS+Sex+Age vs Sex+Age model, then instead of claiming there their results shows "... potential utility of these PRSs in preventative medicine, diagnosis, and other applications", they should instead use PRS models in place of PRSs. This is especially a concern as in line 335, the authors suggested that "It is likely that the PRS of genetic variants contribute less than other covariates".

2. Thank you for the clarification with regard to number of the phenotype and PGS traits pair. Given the context, it might be best to say "1730 phenotype - PGS pairs were identified", instead of "1730 pairs of phenotype with PGS traits", as it still reads like there are 1730 phenotypes.

3. Following the previous point, maybe the authors should standardize their language, either use PGS or PRS, instead of using both interchangeably.

4. It is appreciated that the authors did different sex and age matching procedures to tested the best ratio, it's slightly confusing to read. Given the result, it might be best to only present the 1:8 ratio to keep the main text simply, and put the remaining ratios in the supplementary as a sensitivity analyses.

5. It is highly unusual to not include the PCs in the PGS model. As the PGS is affected by population stratification, which is "controlled" by including the PCs. Personally, I would not trust PGS models without PCs as a covariates. Based on the authors observation, it is also clear that the model with PCs seems to perform the best.

6. Again, if the authors want to argue that the PGS is useful for prediction, then the best comparison will be the compare the PGS model with covariates against the model with only covariates, to see if the PGS improves upon the model. If it does not, then even if the model provide highly significant value, that'd likely only be due to Sex and Age, which are known to predict majority of human disease.

7. I am slightly concern with the correlation analyses mainly because for each traits, there can be at most 5 PGSs (if I read correctly). These PGSs can be highly correlated with each other (e.g. they can be derived from the same GWAS, with slightly different flavor of PGS construction or covariates). The traits can also be highly correlated (e.g. Height and BMI). Given that it is difficult to account for the amount of correlation between the traits and PGSs, we cannot discern the whether the correlation observed between the matching ratio, the number of SNPs and the performance of the PGS model, were true signal, or was driven by a small number of enriched trait. As such, we won't be able to draw a conclusion from these.

8. Were the 52 phenotypes observed with $AUC > 0.6$ a subset of the 63 phenotypes when covariates were included?

9. The S-shaped curve is expected for any PGS that were associated with the phenotype of interest. You can refer to the "Graphical representations of results: bar and quantile plots" section of the tutorial paper: Tutorial: a guide to performing polygenic risk score analyses

10. Table 2 and 3 is best presented as a supplementary.

11. Line 423 might not be completely true, as some PGS does considered the environmental factors by incorporating them as a covariate.

Minor comment:

1. Line 60 can be simplified to "Polygenic Score (PGS) Catalog was developed to facilitate the distribution of PGS [cite]"

2. It is highly appreciated that the authors provide their code. However, not sure word is the best media for that. A better media might be just the Rscript files, bash files or even better, github or gitlab

Manuscript ID: *NCOMMS-23-25041B*

Titled “Applicability of Polygenic Scores for Disease Prediction Across a Wide Range of Disease Traits in a Hospital-Based Cohort”.

As per the suggestions of the referees, we have made changes to our manuscript and take into considerations of the points and questions raised by the referees. The revised contents were marked in red in the revised version of manuscript. We have made the following changes in our manuscript:

1. Replace PRS with PGS in the title and throughout the main article
2. Correct the wording in the Introduction, page 2
3. Use "phenotype - PGS pairs" to clarify the number of phenotype and PGS trait pairs
4. Add the sample cohort distribution for PGS development from PGS Catalog in Figure 4.
5. Analyze the impact of PGS developed by different sample cohort on our results in the Section III, part B
6. Add Age+Sex models in to the article
7. Update all tables and figures based on new results

Response to Reviewer #1

Comments:

In my opinion the authors need to show the effect of the sample sizes (cases and control of the data from which the PGS were derived) on quality of the prediction - at least for the set of PGS that have that information for. My contention is that whatever performance statistics reported in this study will probably reflect on the PGS data – so the inference made based on their results would be misleading considering the quality of the PGS data was not considered. For example, the authors did not clarify why there was change in which trait's PRS performance was best performing - and such instability is concerning.

I do not doubt the quality of curation by the author but if that curation does not address the details on PGS data it gives me pause. Therefore, I believe the size of source data from which PGS were generated is an important parameter to assess the quality of prediction in their data - this impact how the prediction of a trait is assessed - and I am not sure why the authors choose to sidestep it.

Reply to Reviewer #1: Thank you for your thoughtful comments and inquiries. In our study, we explicitly acknowledge the variability in data collection and cleaning procedures across the studies cataloged in the PGS Catalog. To address the concern about the size of the source data used for PGS development, we conducted analyses at different stages. Through an initial screening process, we retained PGS with up to 5 candidates, considering the P-value from the Wilcoxon rank sum test. This screening effectively filtered out PGS traits with smaller sample cohorts. The outcomes of these analyses are detailed in **Section III, part B “Regarding the sample cohort used to develop PGS, a total of 2,153 records were available. Since 60.94% (1,312 records) of the data lacked case/control values, we used the number of individuals for statistical analysis. From the data distribution (Figure 4C), it is observed that 50% of PGS were developed using sample less than 23,072 individuals. As a result of an initial screening step, 507 PGS were retained, and the cumulative distribution plot (Figure 4D) shows that 50% of PGS using sample sizes larger than 269,704. This indicates a tendency in our process to retain PGS with relatively larger sample sizes. Similar results were observed in the final 201 PGS used for optimized models.”** Furthermore, for the selected PGS, we compared their AUC performance in our models. The analysis result is documented in **Supplementary Table 3** and has been integrated into **Section III, part B “The sample cohort used to develop the PGS and our AUC in the PGS model did not show any significant association after an initial screening step (Supplementary table3)”**. According to this result, the AUC of the final analysis is not affected by the number of samples in the PGS developed after the initial screening step.

Response to Reviewer #2:

Comments:

In this updated, Sun et al has addressed most of my previous concern, and I thank the authors for their hard work. Unfortunately, with there are still some unaddressed concerns:

1. If the authors were interested in the PRS model, instead of the performance of the PRS score (as suggested by their reply to my previous Q1), then it'd be best to make it clear in the abstract too. For example, as the authors did not perform tests on the improved performance of PRS over covariates e.g. PRS+Sex+Age vs Sex+Age model, then instead of claiming their results shows "... potential utility of these PRSs in preventative medicine, diagnosis, and other applications", they should instead use PRS models in place of PRSs. This is especially a concern as in line 335, the authors suggested that "It is likely that the PRS of genetic variants contribute less than other covariates".
2. Thank you for the clarification with regard to number of the phenotype and PGS traits pair. Given the context, it might be best to say "1730 phenotype - PGS pairs were identified", instead of "1730 pairs of phenotype with PGS traits", as it still reads like there are 1730 phenotypes.
3. Following the previous point, maybe the authors should standardize their language, either use PGS or PRS, instead of using both interchangeably.
4. It is appreciated that the authors did different sex and age matching procedures to tested the best ratio, it's slightly confusing to read. Given the result, it might be best to only present the 1:8 ratio to keep the main text simply, and put the remaining ratios in the supplementary as a sensitivity analyses.
5. It is highly unusual to not include the PCs in the PGS model. As the PGS is affected by population stratification, which is "controlled" by including the PCs. Personally, I would not trust PGS models without PCs as a covariates. Based on the authors observation, it is also clear that the model with PCs seems to perform the best.
6. Again, if the authors want to argue that the PGS is useful for prediction, then the best comparison will be the compare the PGS model with covariates against the model with only covariates, to see if the PGS improves upon the model. If it does not, then even if the model provide highly significant value, that'd likely only be due to Sex and Age, which are known to predict majority of human disease.
7. I am slightly concern with the correlation analyses mainly because for each traits, there can be at most 5 PGSs (if I read correctly). These PGSs can be highly correlated with each other (e.g. they can be derived from the same GWAS, with slightly different flavor of PGS construction or covariates). The traits can also be highly correlated (e.g. Height and BMI). Given that it is difficult to account for the amount of correlation between the traits and PGSs, we cannot discern the whether the correlation observed between the matching ratio, the number of SNPs and the performance of the PGS model, were true signal, or was

driven by a small number of enriched trait. As such, we won't be able to draw a conclusion from these.

8. Were the 52 phenotypes observed with $AUC > 0.6$ a subset of the 63 phenotypes when covariates were included?
9. The S-shaped curve is expected for any PGS that were associated with the phenotype of interest. You can refer to the "Graphical representations of results: bar and quantile plots" section of the tutorial paper: Tutorial: a guide to performing polygenic risk score analyses
10. Table 2 and 3 is best presented as a supplementary.
11. Line 423 might not be completely true, as some PGS does considered the environmental factors by incorporating them as a covariate.

Minor comments:

1. Line 60 can be simplified to "Polygenic Score (PGS) Catalog was developed to facilitate the distribution of PGS [cite]"
2. It is highly appreciated that the authors provide their code. However, not sure word is the best media for that. A better media might be just the Rscript files, bash files or even better, github or gitlab.

Reply to Reviewer #2:

Q1 If the authors were interested in the PRS model, instead of the performance of the PRS score (as suggested by their reply to my previous Q1), then it'd be best to make it clear in the abstract too. For example, as the authors did not perform tests on the improved performance of PRS over covariates e.g. PRS+Sex+Age vs Sex+Age model, then instead of claiming there their results shows "... potential utility of these PRSs in preventative medicine, diagnosis, and other applications", they should instead use PRS models in place of PRSs. This is especially a concern as in line 335, the authors suggested that "It is likely that the PRS of genetic variants contribute less than other covariates".

Reply for Q1: Thank you for your valuable advice. We have incorporated the Sex+Age model into the current version of our article. Upon careful observation, we indeed noted instances where the model performance for certain phenotypes was mainly contributed by the inclusion of Sex+Age.

The updated results reflecting the performance of the Sex+Age model have been incorporated into **Section III, from parts B to E** of the article. Additionally, these results are now accessible on the GeneAnaBase website (<https://pgscatalog.azure.nihxcmuh.org/#/>) for comprehensive visibility.

Q2 Thank you for the clarification with regard to number of the phenotype and PGS traits pair. Given the context, it might be best to say "1730 phenotype - PGS pairs were identified", instead of "1730 pairs of phenotype with PGS traits", as it still reads like there are 1730 phenotypes.

Reply for Q2: Thank you for your valuable advice. We have comprehensively addressed potential confusion by modifying the wording throughout the article.

Q3 Following the previous point, maybe the authors should standardize their language, either use PGS or PRS, instead of using both interchangeably.

Reply for Q3: Thank you for your valuable suggestion. Recognizing the potential for confusion with the interchangeability of PGS and PRS in our language usage, we have revised the entire manuscript. In accordance with your advice, we have consistently replaced 'PRS' with 'PGS' (polygenic score) throughout the article.

Q4 It is appreciated that the authors did different sex and age matching procedures to tested the best ratio, it's slightly confusing to read. Given the result, it might be best to only present the 1:8 ratio to keep the main text simply, and put the remaining ratios in the supplementary as a sensitivity analyses.

Reply for Q4: Thank you for your valuable suggestion. We have streamlined the presentation of our results by exclusively showcasing the 1:8 ratio in the main text. The comprehensive set of results, including additional ratios, is now available on the GeneAnaBase website (<https://pgscatalog.azure.nihxcmuh.org/#/>).

Q5 It is highly unusual to not include the PCs in the PGS model. As the PGS is affected by population stratification, which is "controlled" by including the PCs. Personally, I would not trust PGS models without PCs as a covariates. Based on the authors observation, it is also clear that the model with PCs seems to perform the best.

Reply for Q5: Taiwan falls within the East Asian region and can be further subdivided into five distinct populations: Japanese in Tokyo (JPT), Han Chinese in Beijing (CHB), Southern Han Chinese (CHS), Chinese Dai in Xishuangbanna (CDX), and Kinh in Ho Chi Minh City (KHV). Through data obtained from the Million Person Precision Medicine Initiative at China Medical University Hospital, we discerned that the CMUH database is primarily composed of individuals with Southern Han Chinese ancestry, with a minority representation of individuals from CHB and CDX populations. Given this genetic composition, the inclusion of PCs in our study may slightly enhance the AUC when added to age, sex, and PGS.

Q6 Again, if the authors want to argue that the PGS is useful for prediction, then the best comparison will be the compare the PGS model with covariates against the model with only covariates, to see if the PGS improves upon the model. If it does not, then even if the model

provide highly significant value, that'd likely only be due to Sex and Age, which are known to predict majority of human disease.

Reply for Q6: As mentioned in response to Q1, we have incorporated the Sex+Age model into our analysis. Upon careful observation, we indeed noted instances where the model performance for certain phenotypes is influenced by Sex+Age, suggesting that the predictive capacity provided by PGS may be limited. The comprehensive set of updated results, including the Sex+Age model, is now available in **Section III, parts B to E** of the article and on the GeneAnaBase website (<https://pgscatalog.azure.nihxcmuh.org/#/>).

Q7 I am slightly concern with the correlation analyses mainly because for each traits, there can be at most 5 PGSs (if I read correctly). These PGSs can be highly correlated with each other (e.g. they can be derived from the same GWAS, with slightly different flavor of PGS construction or covariates). The traits can also be highly correlated (e.g. Height and BMI). Given that it is difficult to account for the amount of correlation between the traits and PGSs, we cannot discern the whether the correlation observed between the matching ratio, the number of SNPs and the performance of the PGS model, were true signal, or was driven by a small number of enriched trait. As such, we won't be able to draw a conclusion from these.

Reply for Q7: In our analysis, we systematically compared models by retaining only the most optimized PGS, matching method, and covariate inclusion conditions for each phenotype. Admittedly, as you pointed out, the lack of a standardized and objective method for evaluating PGS scoring formulas poses a challenge. Our approach relies on observing the associations between the currently optimized models and various measurable outcomes in our dataset.

We recognize the need for a more rigorous and objective evaluation of PGS scoring formulas, and we acknowledge the limitations of our current methodology. The observed correlations between the matching ratio, number of SNPs, and the performance of the PGS model are, indeed, influenced by the chosen conditions and the inherent complexities of predictive modeling.

As highlighted in our discussion, we are cognizant of these challenges and are committed to addressing them in future research. We believe that refining our approach, possibly through the exploration of more sophisticated statistical methods and collaboration with experts in the field, will contribute to a more comprehensive understanding of PGS scoring formulas. The intricacies of disease prediction models, especially in the context of varying disease prevalence rates, are factors we aim to further investigate for improved evaluation in subsequent studies. We appreciate your valuable insights and recognize the evolving nature of this research field.

Q8 Were the 52 phenotypes observed with $AUC > 0.6$ a subset of the 63 phenotypes when covariates were included?

Reply for Q8: No, the two conditions are independent. The primary emphasis is on conveying that an increase in the number of included covariates corresponds to a higher count of phenotypes with $AUC > 0.6$. To avoid confusion, we have split the information into four sentences: **”When we set a threshold of $AUC > 0.6$ to signify effective model performance, we observed an increase in the number of phenotypes surpassing this threshold as more covariates were incorporated. For the models trained with age and sex, 24 phenotypes achieved an $AUC > 0.6$. For the model trained with PGS alone, 26 phenotypes achieved an $AUC > 0.6$. For the model trained with PGS, age and sex, 47 phenotypes were observed. For the model trained with PGS, age, sex, and the first four principal components, 47 phenotypes were observed.“**

Q9 The S-shaped curve is expected for any PGS that were associated with the phenotype of interest. You can refer to the "Graphical representations of results: bar and quantile plots" section of the tutorial paper: Tutorial: a guide to performing polygenic risk score analyses

Reply for Q9: While it is true that a well-performing PGS scoring formula is expected to exhibit an S-shaped curve, our study is centered on practical applications. Consequently, PGS scoring formulas designed in different studies may not necessarily be suitable for East Asian populations or the specific dataset we have selected for individuals with the same phenotype. This can result in non-S-shaped numerical distributions or poor AUC values. We elaborate on these observations in **Section III part D** of the article, recognizing the need to consider the applicability of PGS scoring formulas across diverse populations and datasets.

Q10 Table 2 and 3 is best presented as a supplementary.

Reply for Q10: After consolidating the presentation of only the 1:8 ratio in Table 2, the Table 3 has been removed. This decision stems from the comprehensive availability of our results on the GeneAnaBase website (<https://pgscatalog.azure.nihxcmuh.org/#/>), eliminating the need for redundant information in the document.

Q11 Line 423 might not be completely true, as some PGS does considered the environmental factors by incorporating them as a covariate.

Reply for Q11: It is true that the PGS Catalog incorporates environmental factors in the design and evaluation of AUC. The initial statement might lead to misunderstanding, as such, we have revised it to: **”Furthermore, although the PGS Catalog includes models including environmental factors or gene–environment interactions, users can only obtain PGS scoring formula without the impact/weight related to environmental factors, which could complicate model application and reduce AUC”**

Minor Q1. Line 60 can be simplified to "Polygenic Score (PGS) Catalog was developed to facilitate the distribution of PGS [cite]"

Reply for Minor Q1: Thank you for your suggestion. We have corrected the wording in the introduction section. **“Polygenic Score (PGS) Catalog (<https://www.pgscatalog.org/>) [21] was developed to facilitate the distribution of PGS. This catalog adheres to standardized procedures for quality control, data curation, and metadata annotation, serves as a centralized resource for researchers and clinicians to access and use PGSs for various applications, such as risk prediction, personalized medicine, and genetic research.”**

Minor Q2. It is highly appreciated that the authors provide their code. However, not sure word is the best media for that. A better media might be just the Rscript files, bash files or even better, github or gitlab.

Reply for Minor Q2: We appreciate your suggestion, and we are currently in the process of developing an automated tool to package and share our code with a broader audience. This initiative involves overcoming various challenges, such as the code spanning different languages (R, Python, Linux) and requiring manual updates. Once development is complete, we plan to promptly publish the code on GitHub and provide comprehensive technical documentation to assist users. We understand the importance of accessibility and are working diligently to streamline the sharing process.

REVIEWER COMMENTS

Reviewer #1 (Remarks to the Author):

Authors have addressed the issue I raised and I am satisfied with their response

Reviewer #2 (Remarks to the Author):

In this revision, the authors have addressed several concerns raised in my previous comments. However, I still find it challenging to discern the primary message of the paper. While I have no intention of delaying the paper's publication unnecessarily, and I am open to the editor proceeding with publication, I believe there is untapped potential in the data and analyses. This potential may not have been fully realized due to the manner in which the results and methods have been presented. Specifically, the data presented here, the CMUH, is a unique and valuable dataset comprise mainly of East Asians, and it would be of great interest to investigate how existing PGS applies in this dataset, and I believe was the authors' original intention. However, there are several issues with the presentations that need to be addressed:

1. The analyses are un-necessarily complicated. For one, as mentioned in my previous comments, there is no need to discuss the case control matching in such detail in the main text. The presentation raised more questions than answers and were confusing to read. For one, how was those P-threshold determined? A simpler approached would be to simply use the 8x matching and put the remaining matching results to the supplementary, as suggested before. Same for figure 3.
2. While it might be of interest to the authors to test different covariate models, it dilutes the message, and lead to confusion to readers as it is unclear if the authors were trying to determine the performance of the PGS models in CMUH, or if they were trying to investigate the added value of PGS on existing models. A much simpler approach would be to only consider two models, the PGS model with Sex, age and PCs, and the Non-PGS model with only Sex, Age and PCs.
3. It is also confusing as to why the authors would use the Wilcoxon rank sum test for their main results. The p-value of the full PGS model form the logistic regression (which the authors performed to obtained the AUC), or the p-value of the PGS within the full model, would both be better candidate for presentation. In addition, if the authors would like to present the added performance of PGS, then they can use the likelihood ratio test on the PGS and non-PGS model, using `lrtest`` function in R (<https://www.statology.org/likelihood-ratio-test-in-r/>). Which would be more interesting.
4. Figure 4 is more suitable maybe for a review for the PGS catalog and does not really help with the content of the current paper.
5. Alternatively, it'd also be interesting to see how the AUC / R2 compared to the EUR AUC / R2 presented in PGS catalog. For example, replace Figure 4 with bar charts of the EUR AUC presented in the PGS catalog (only for traits with this information), and next to the PGS catalog's performance, show the performance in CMHU. Alternatively show the relative difference between the two.
6. In addition, due to the data structure, I do not believe the correlation analyses is correct as there is an over-representation of traits that were well studied (e.g. cardiovascular) and would dominate the correlation, leading to biased interpretation. Number of SNPs included in the calculation is also more of the measurement of method used, rather than the property of the score, as some methods (e.g. LDpred) will include all SNPs in the calculation, but with a lot of those SNPs having an effective size close to or equal to zero, rendering this measurement meaningless.
7. If observed closely, in supplementary table 3, we can note that a lot of the PGS have the same number of individuals, as a lot of the PGS records are from UKB.
8. For supplementary table 3, it appears that many correlation tests were performed based on less than 5 records, which might not be sufficient to draw robust conclusions.
9. The confidence interval for all PGSID being the same within the same phenotype could be clarified. If these CI were observed in CMUH, a more detailed table legend will be more helpful.
10. Side note: good candidate to generate pipeline would be `nextflow`` or `snakemake``.

Reviewer #2 (Remarks on code availability):

While the codes are available, it'd be rather challenging for anyone to use it considering that it is presented in a word document.

Manuscript ID: *NCOMMS-23-25041C*

Titled “Applicability of Polygenic Scores for Disease Prediction Across a Wide Range of Disease Traits in a Hospital-Based Cohort”.

As per the suggestions of the referees, we have made changes to our manuscript and take into considerations of the points and questions raised by the referees. The revised contents were marked in red in the revised version of manuscript. We have made the following changes in our manuscript:

1. Change the title to “Polygenic Scores Utility across Diverse Diseases in a Hospital Cohort for Predictive Modeling”
2. Move the “The effect of case control matching ratio on the statistical value of PGS distribution” part to the Supplementary Results.
3. Rearrange the figure order of Figures 2–6.
4. Add a new subplot in Figure 3.

Response to Reviewer #2:

Comments:

In this revision, the authors have addressed several concerns raised in my previous comments. However, I still find it challenging to discern the primary message of the paper. While I have no intention of delaying the paper's publication unnecessarily, and I am open to the editor proceeding with publication, I believe there is untapped potential in the data and analyses. This potential may not have been fully realized due to the manner in which the results and methods have been presented. Specifically, the data presented here, the CMUH, is a unique and valuable dataset comprise mainly of East Asians, and it would be of great interest to investigate how existing PGS applies in this dataset, and I believe was the authors' original intention. However, there are several issues with the presentations that need to be addressed:

1. The analyses are un-necessarily complicated. For one, as mentioned in my previous comments, there is no need to discuss the case control matching in such detail in the main text. The presentation raised more questions than answers and were confusing to read. For one, how was those P-threshold determined? A simpler approached would be to simply use the 8x matching and put the remaining matching results to the supplementary, as suggested before. Same for figure 3.
2. While it might be of interest to the authors to test different covariate models, it dilutes the message, and lead to confusion to readers as it is unclear if the authors were trying to determine the performance of the PGS models in CMUH, or if they were trying to investigate the added value of PGS on existing models. A much simpler approach would be to only consider

two models, the PGS model with Sex, age and PCs, and the Non-PGS model with only Sex, Age and PCs.

3. It is also confusing as to why the authors would use the Wilcoxon rank sum test for their main results. The p-value of the full PGS model from the logistic regression (which the authors performed to obtain the AUC), or the p-value of the PGS within the full model, would both be better candidates for presentation. In addition, if the authors would like to present the added performance of PGS, then they can use the likelihood ratio test on the PGS and non-PGS model, using the `lrtest` function in R (<https://www.statology.org/likelihood-ratio-test-in-r/>). Which would be more interesting.

4. Figure 4 is more suitable maybe for a review for the PGS catalog and does not really help with the content of the current paper.

5. Alternatively, it'd also be interesting to see how the AUC / R² compared to the EUR AUC / R² presented in PGS catalog. For example, replace Figure 4 with bar charts of the EUR AUC presented in the PGS catalog (only for traits with this information), and next to the PGS catalog's performance, show the performance in CMHU. Alternatively show the relative difference between the two.

6. In addition, due to the data structure, I do not believe the correlation analyses is correct as there is an over-representation of traits that were well studied (e.g. cardiovascular) and would dominate the correlation, leading to biased interpretation. Number of SNPs included in the calculation is also more of the measurement of method used, rather than the property of the score, as some methods (e.g. LDpred) will include all SNPs in the calculation, but with a lot of

those SNPs having an effective size close to or equal to zero, rendering this measurement meaningless.

7. If observed closely, in supplementary table 3, we can note that a lot of the PGS have the same number of individuals, as a lot of the PGS records are from UKB.

8. For supplementary table 3, it appears that many correlation tests were performed based on less than 5 records, which might not be sufficient to draw robust conclusions.

9. The confidence interval for all PGSID being the same within the same phenotype could be clarified. If these CI were observed in CMUH, a more detailed table legend will be more helpful.

10. Side note: good candidate to generate pipeline would be `nextflow` or `snakemake`.

Reply to Reviewer #2:

Q1 The analyses are un-necessarily complicated. For one, as mentioned in my previous comments, there is no need to discuss the case control matching in such detail in the main text. The presentation raised more questions than answers and were confusing to read. For one, how was those P-threshold determined? A simpler approached would be to simply use the 8x matching and put the remaining matching results to the supplementary, as suggested before. Same for figure 3.

Reply for Q1: We appreciate your valuable advice. We have relocated the case-control matching results to the Supplementary Results section for better organization. For the determination of the P-threshold, our primary reference was the paper available at

<https://doi.org/10.1371/journal.pgen.1010105>. Your feedback is greatly appreciated, and the adjustments made aim to enhance clarity by simplifying the presentation.

Q2 While it might be of interest to the authors to test different covariate models, it dilutes the message, and lead to confusion to readers as it is unclear if the authors were trying to determine the performance of the PGS models in CMUH, or if they were trying to investigate the added value of PGS on existing models. A much simpler approach would be to only consider two models, the PGS model with Sex, age and PCs, and the Non-PGS model with only Sex, Age and PCs.

Reply for Q2: Your advice is invaluable. Our goal is to compare our results with those from the PGS catalog models. Given the diverse covariate conditions present in many studies, we believe that comparing multiple conditions simultaneously can deepen our understanding of the information. As observed in Figure 4, some models achieve relatively higher AUCs when using PGS alone. Limiting the comparison to age + sex and age + sex + PGS might omit some valuable discussion topics.

Q3 It is also confusing as to why the authors would use the Wilcoxon rank sum test for their main results. The p-value of the full PGS model from the logistic regression (which the authors performed to obtain the AUC), or the p-value of the PGS within the full model, would both be better candidate for presentation. In addition, if the authors would like to present the added performance of PGS, then they can use the likelihood ratio test on the PGS and non-PGS model, using `lrtest` function in R (<https://www.statology.org/likelihood-ratio-test-in-r/>). Which would be more interesting

Reply for Q3: The Wilcoxon rank sum test and logistic regression serve different purposes in

our study. The Wilcoxon rank sum test is used in the initial step, before model training, to determine if a statistically significant difference exists between two groups. Following this, a logistic regression model is built to predict the probability of a binary outcome based on predictor variables. As AUC is commonly used for model comparison in many studies (as mentioned in Fig. 3), we chose to present our results using AUC.

Q4 Figure 4 is more suitable maybe for a review for the PGS catalog and does not really help with the content of the current paper.

Reply for Q4: We appreciate your valuable suggestions. Our aim is to compare our results with those from the PGS catalog. We believe that this figure can effectively communicate to readers that the PGS Catalog predominantly uses AUC as the evaluation metric for PGS models, covering various data types and particularly focusing on the European population. We trust that this information will assist readers in understanding our results.

Q5 Alternatively, it'd also be interesting to see how the AUC / R2 compared to the EUR AUC / R2 presented in PGS catalog. For example, replace Figure 4 with bar charts of the EUR AUC presented in the PGS catalog (only for traits with this information), and next to the PGS catalog's performance, show the performance in CMHU. Alternatively show the relative difference between the two.

Reply for Q5: We value your suggestion. As mentioned in the previous response, we have included the distribution of AUC across different sample cohorts, providing readers with additional information. This, in conjunction with Table 1, is made available for reference.

Q6 In addition, due to the data structure, I do not believe the correlation analyses is correct as there is an over-representation of traits that were well studied (e.g. cardiovascular) and would

dominate the correlation, leading to biased interpretation. Number of SNPs included in the calculation is also more of the measurement of method used, rather than the property of the score, as some methods (e.g. LDpred) will include all SNPs in the calculation, but with a lot of those SNPs having an effective size close to or equal to zero, rendering this measurement meaningless.

Reply for Q6: Indeed, different methods can introduce statistical biases. However, we have mitigated this by removing zero-value points from the score file during variant count calculations. Furthermore, we have capped the variant count at 14,029,683, the maximum obtainable from chip analysis. As indicated in Supplementary Table 2, each phenotype undergoes an average of 250 PGS scoring tests, encompassing various SNP selection methods. Before conducting correlation analyses, only a unique PGS is retained from the Wilcoxon rank sum test and logistic regression tests for subsequent comparisons. Therefore, if LDpred stands out in the end, we use the calculated number of obtainable variants, making the results reliable.

Q7 If observed closely, in supplementary table 3, we can note that a lot of the PGS have the same number of individuals, as a lot of the PGS records are from UKB.

Reply for Q7: Thank you for your thorough observation. Indeed, as you mentioned, PGS scoring tests on average take 250 per phenotype, with larger studies like UKB standing out.

Q8 For supplementary table 3, it appears that many correlation tests were performed based on less than 5 records, which might not be sufficient to draw robust conclusions.

Reply for Q8: The results of case-control matching have been moved to the Supplementary Results. This supplementary information does not need to remain in the main text, and therefore, we have removed this section.

Q9 The confidence interval for all PGSID being the same within the same phenotype could be clarified. If these CI were observed in CMUH, a more detailed table legend will be more helpful.

Reply for Q9: Considering the majority of entries in the PGS Catalog from European populations and the diverse use of covariates, as mentioned earlier, the inclusion of disease-related environmental factors could significantly bolster the AUC. However, the absence of PGS performance data specific to individual studies within the PGS Catalog poses a challenge for comparing the efficacy of PGS in isolation.

Q10 Side note: good candidate to generate pipeline would be `nextflow` or `snakemake`

Reply for Q10: Thank you for your suggestion. We acknowledge that both Nextflow and Snakemake are new and appropriate tools for our purposes. We have already uploaded the existing code, which is currently integrated with Linux, onto GitHub for user accessibility. If transitioning to these tools can improve efficiency, we will work on transitioning as soon as possible.